# Rationally designed laterally-condensed-catalysts deliver robust activity and selectivity for ethylene production in acetylene hydrogenation

Future carbon management strategies require storage in elemental form, achievable through a sequence of $CO_2$ hydrogenation reactions. Hydrogen is recycled from molecular intermediates by dehydrogenation, and side product acetylene selectively hydrogenated to ethylene. Existing Pd alloy catalysts for gas purification underperform in concentrated feeds, necessitating novel concepts. Atomistic simulations unveil superior selectivity of Pd:C solid solutions that optimize chemisorption energies and preclude sub-surface hydrides, verified here with model thin films. Multiple design criteria deduced from conventional catalysts facilitate synthesizing a self-repairing Pd:C system of a laterally condensed catalyst (LCC). A Pd layer prepared on a designated $SiO_2$ buffer layer enables control of reactive interface, sub-surface volume and extended functional interface towards the buffer. Function and metric are supervised by operando micro-spectroscopy. This catalyst design shows, ethylene productivity >1 $kmol_{C2H4}/g_{Pd}$/hour is reproducibly achieved and benchmarked against known catalysts. Photovoltaics deposition technologies enable scalability on real-world substrates saving active metal. A design-of-experiment approach demonstrates the improvement potential of the LCC approach.

The transformation of energy systems to mitigate greenhouse gas emissions will contain an element of carbon management[1]. $CO_2$ from inevitable sources or direct air capture and low-carbon hydrogen will be converted to platform molecules[2,3] for the chemical industry, to e-fuels and to solid carbon. The latter can be deposited[4] to guarantee that the molecular products are carbon-negative or carbon neutral. A tentative flow chart of such a process is shown in Fig. 1 (Supplementary Note 1). Chemical reduction of $CO_2$ while co-generating valuable basic chemicals must preserve hydrogen as much as possible. Such reactions leading to black carbon will co-generate ethylene and acetylene by plasma pyrolysis. The highly concentrated acetylene fraction[5] should be selectively hydrogenated to valuable ethylene to eliminate operational dangers of the desired low-temperature carbon formation.

In concentrated gas streams, the exothermic semi-hydrogenation[6] is more demanding than the thermodynamically preferred full hydrogenation and thus, the formation of polymeric carbon must be suppressed (Supplementary Note 1).

Selective hydrogenation of diluted acetylene to ethylene is a long-practiced reaction to purify fossil ethylene streams[7]. To achieve rapid and selective production of ethylene, the reaction conditions and catalysts must operate far from equilibrium. Diluted gas streams, high temperatures and Pd catalysts with damped full hydrogenation activity (by adding metals, CO or carbonaceous overlayers)[7,8] are used[9] in purification applications. Such conditions are unsuitable (Supplementary Table 1) for large-scale energy-related applications indicated in Fig. 1.

✉e-mail: skorupska@fhi-berlin.mpg.de; rs01@fhi-berlin.mpg.de

**Fig. 1 | A process that converts CO$_2$ simultaneously to solid carbon and to feedstock molecules for the chemical industry.** Renewable energy is used to formally revert the combustion process (separating carbon from oxygen). Two separate hydrogenation steps for CO$_2$ to methanol and hydrocarbons are used for robust operation of the combined process using known procedures[62] that can cope with the complex feedstock. A critical step past the treatment of the residual gaseous mixture is the safe removal of acetylene (magenta step).

A completely new catalysis design concept, suitable for rapid and reliable conversion in high-volume gas streams, seems thus adequate, especially taking into account the dynamic nature of the modifying H and C elements in Pd catalysts[10] that is often not considered under mild[11,12] reaction conditions. Supported modified Pd catalysts with mechanochemical synthesis have been suggested[13]. Combination of adapted particle morphology and strong metal-support interaction induced by the mechanochemical activation of the support oxide was found to be essential for stable high performance in concentrated acetylene semi-hydrogenation.

In the following, we describe the use of prior knowledge to derive a design concept for a catalyst suitable for carbon management applications (Supplementary Note 1). Then we illustrate its realization using a rational approach including model systems (Supplementary Note 3) and ample analytical control (Supplementary Notes 5, 6, 8).

Pathfinding experiments to design the buffer layer are described in Supplementary Note 4. We used theory to guide the development (Supplementary Note 2). A large number of samples as thin films and for reference in other forms were prepared and used here as indicated in the sample table in Supplementary Information. Performance data (Supplementary Notes 7 and 9) were benchmarked experimentally and against multiple literature data (Supplementary Tables 1, 7 and 8). The design of experiment approach carried out is illustrated in Supplementary Note 9.

## Results and discussion

### Concept development

Prior knowledge[9,14] on this reaction allows formulating design criteria for catalysts. This is critical to provide a robust and scalable catalyst material for performing the carbon management reaction sequence of Fig. 1 in the volume required for a relevant contribution to climate protection. Structural dynamics of the reactive interface are needed to provide the active sites[15]. It is enabled as rough nano-morphology[14,16] to minimize condensation reactions and the dissolution of hydrogen[17]. Robust chemical interaction provides structural stability at the functional interface between the catalyst and its support. The support must remain neutral to reagents and products and thus be free of acid-base reactivity. Extended terraces on Pd of defined surface orientation[18] are unlikely to result in a stable catalyst with low tendency to become blocked by side products. Achieving selectivity to ethylene requires avoiding β-PdH formation[7], the segregation of which is responsible for deep hydrogenation[19]. At the intermediate reaction temperatures that are needed for product desorption and to make ethane formation less likely for thermodynamic reasons, the metal hydride formation is spontaneous[20]. It is thus necessary to prevent its formation by blocking the octahedral voids[7,20] in the Pd lattice. For this, either unreactive metal alloys (Ag, Au, Pb) or non-metals like carbon[21] or sulphur[22] have been used. The alloy formation shifts the Pd d-band by geometric site isolation, by changing the interatomic distances, by inducing strain-related restructuring and by bonding interactions with the alloying element[10]. This affects the chemisorption energies of acetylene and ethylene[14,22,23]. Lower acetylene adsorption energies reduce the rate of its hydrogenation, and lower adsorption energies of ethylene protect it from deep hydrogenation or polymerisation. The price to be paid is limited stability, as at relevant partial pressures of hydrogen the hydride formation leads to segregation of the alloying element that loses its moderating function and covers active Pd sites.

Pd may be modified by non-metal atoms, as demonstrated for Pd-B[24] and Pd-S systems[22]. Carbon from sacrificial dehydrogenation of reactants[25] can be used with the advantage of constituting a continuous source of an alloying element that results in a transient suppression of β-PdH formation. Creating a steady state between desired sub-surface site blocking[26] and unavoidable segregation/methanation provides the function of self-healing of the dynamic catalyst in the working state and seems to be a good compromise in such a complex landscape of limiting performance descriptors.

Elements of solid-state kinetics are also required in the design strategy. One is to control by nano structuring[16] the competition in the dissolution/segregation kinetics between H and C. Small particles exhibit surface strain affecting the exchange of atoms between surface and sub-surface, resulting in less[17] sub-surface hydrogen formation and helping the generation of surface roughness. Hydrogen-induced segregation of alloys can be counteracted by executing strain on the particle through a strong metal-support interaction[14] and binding the Pd nanoparticle (NP) to a reactive defect[27] in the support material. If β-PdH reaches the functional interface, the hydrogen may reduce the support and cause sintering or dissolution of support atoms into Pd. This may be counteracted by minimizing the sub-surface volume such that the strain from the functional interface reaches through the entire Pd NP and reduces hydrogen dissolution. The use of single atom

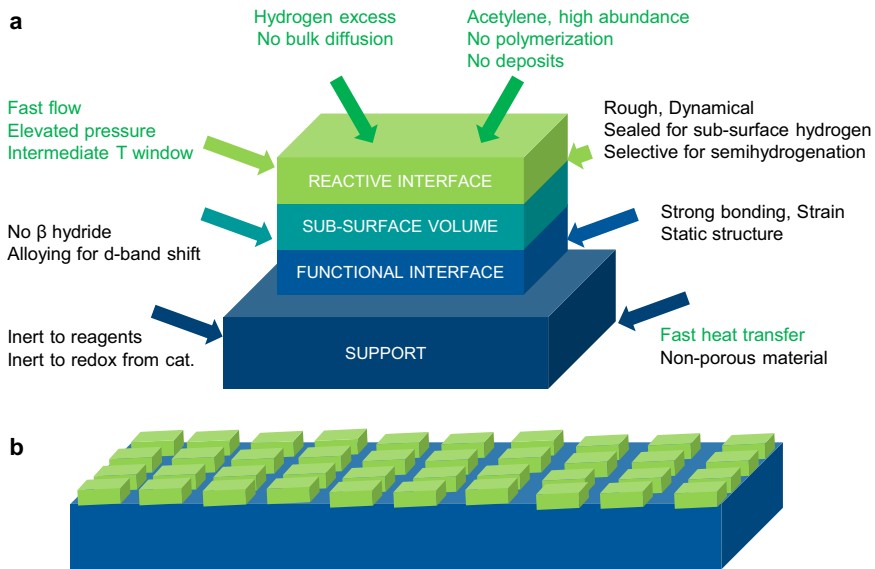

**Fig. 2 | Concept of LCC catalyst. a** Schematic representation of the nanostructured Pd catalyst supported on a Si model support. The arrows list some critical design factors. Coloured factors refer to reagents and conditions, black factors to the catalyst material. We envisage each nanostructure as a tri-layer system with two interfaces. The mesoscopic structure as laterally condensed catalyst (LCC) is given in scheme (**b**). Such self-organization is accessible as the nucleation state of a thin film. Note that we consider parameters at various scales in addition to the local scale of the electronic structure of active sites.

catalysts[9,28,29] is limited by either too strong interaction with the support or by their limited stability during reaction.

We opted for using the dynamic modification approach by carbon[26,30,31] on suitably designed metastable Pd nanostructures realized as nuclei of a thin film with controlled interaction to a tailored innocent SiO₂ support. Figure 2 indicates the non-porous system we intend to generate together with selected design criteria. The chosen "pancake" shape results in a functional interface of the same area than the reactive interface and in a controlled sub-surface volume thickness. Modern thin film technology was chosen for creating metastable systems combined with the capacity for producing large areas of functional materials in technical dimensions and for only applying minimal amounts of active material. The result is a novel from of performance catalyst designated as laterally condensed catalysts (LCC).

Real working catalysts comprise four topologically different regions cooperating during active operation, namely, the top reactive interface, the underlying bulk of the same material, a functional interface and the support of a different material. Existing fundamental studies of catalysts have mainly considered a single reactive interface, largely ignoring the sub-surface and functional interface in contact with the support. This is in part due to the lack of sufficiently sensitive analytics or the inaccessibility of those methods to investigate buried interfaces in nanostructures. Due to its 2D layered planar design, the LCC materials platform proposed here contributes to overcome such challenges.

In the LCC concept, the needed complexity of the reactive and functional interfaces present in nanoparticulate materials is retained. Nonetheless, its planar 2D-like but also highly densely-packed structure closes the materials gap existing between model single crystal catalysts (with one reactive interface and the bulk of the same material), model monolayer thin films (with the reactive interface as a monolayer and the support made of a different material), and the standard 3D nanoparticulate powder industrial catalysts, the latter with the same four different regions and reactive and functional interfaces described above for the LCCs but harder to access and harder to design and stabilize. This planar 2D LCC catalyst platform serves as an intermediate stage between powder catalysts and single-crystal model systems, enabling quantitative fundamental in-situ/operando spectro-microscopy studies and catalytic investigations.

We replaced here the traditional thin films by a laterally-condensed catalyst layer made of closely packed catalyst islands, having in mind the minimization and control of subsurface regions. Such materials concept is not only relevant to thermal catalysis processes but will also have exciting applications in the field of electrocatalysis, where the planar sample geometry is already being applied. The LCCs will in fact allow us to get a holistic understanding of catalytic processes by using the exactly same materials system exposed to two different activation sources (temperature vs. electrical potential).

## Theory and model studies

Density Functional Theory (DFT) confirms the concept of dissolved carbon as modifier for the Pd activity[32,33]. The scaling relation (for calculation details see Methods) presented in Fig. 3a demonstrates the trade-off between activity vs. selectivity. New insights relate to the quantification of the abundance and energetics of spontaneous carbon dissolution into the Pd surface lattice. Its driving force is the chemical interaction between carbon and Pd (Supplementary Note 2), which leads to a weaker surface binding of reaction intermediates. As a result of the observed *d*-band shift, more carbon decreases the acetylene hydrogenation activity and increases the selectivity to ethylene as it is easier for weaker bound C₂H₄ to leave the surface[32,33]. A qualitatively similar effect is expected from alloying Pd with a noble metal such as Ag[33]. Figure 3a suggests increasing selectivity from pure Pd to Pd-Ag and to C-intercalated Pd as a direct function of the carbon concentration. Substantial filling of the Pd lattice with carbon is thermodynamically favorable and the Pd:C system is self-repairing. Supplementary Fig. 1 predicts that the energy gain for carbon dissolution vanishes only just above 1/3 filling of the octahedral voids at the Pd(111) subsurface. Theory thus supports the concept that a spontaneous reaction between Pd and C can create, in a self-limited fashion, the optimal compromise between activity and selectivity. This motivates us to experimentally prepare first a model catalyst with a well-controlled subsurface carbon in a clean UHV system and later a benchmark system which can be realized in a practical environment.

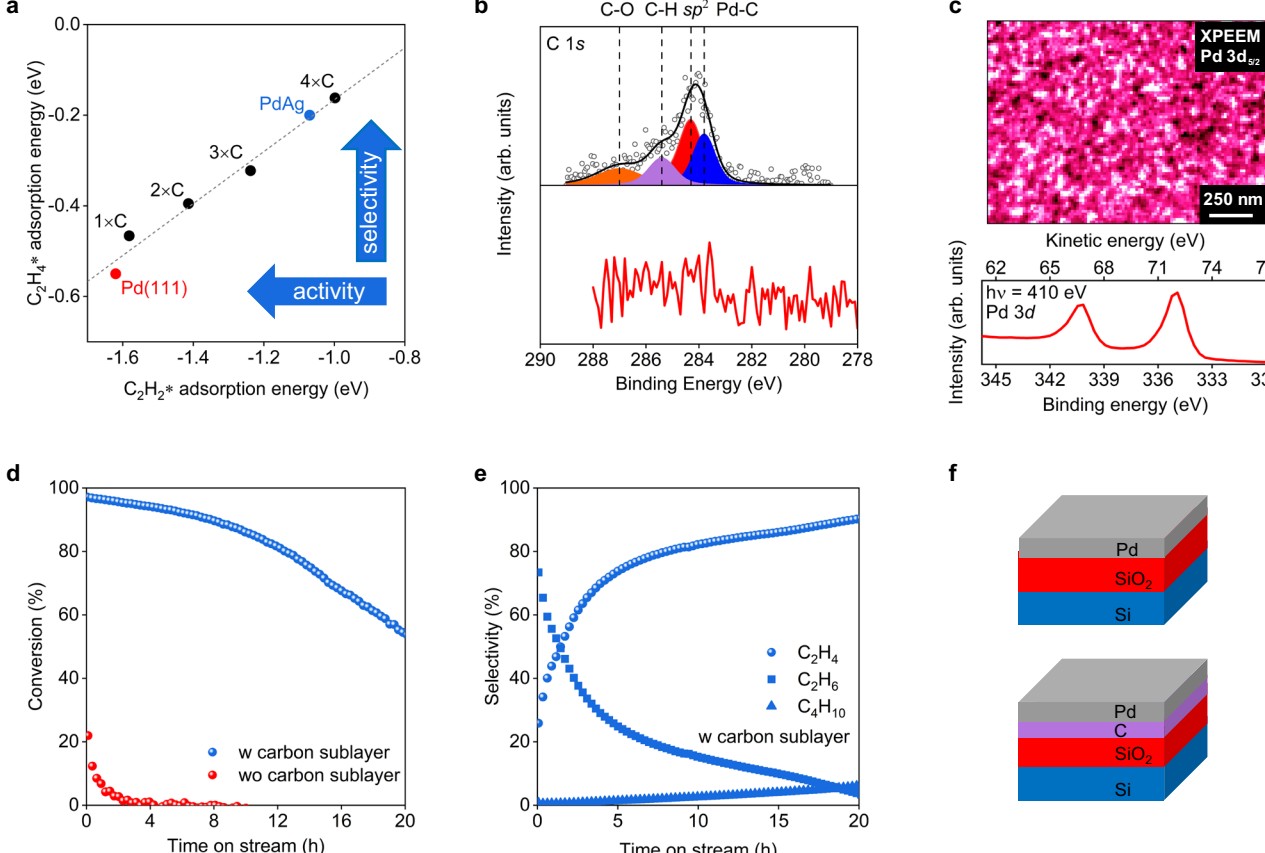

**Fig. 3 | Theory and model experiments. a** Scaling relation between DFT-calculated adsorption energies of $C_2H_2$ and $C_2H_4$: on a clean Pd(111) surface (red), and with increasing concentration of subsurface C atoms within the first interstitial layer (black). Model of a Pd-Ag alloy is also represented for comparison (blue). The number of added C atoms is reported per $3 \times 3 \times 4$ simulation cell as described in the methods. The dotted line is a linear fit to guide the eye. **b** C 1$s$ measured by XPS for LCC Pd with (top spectrum) and without carbon sublayer (bottom spectrum). The signal at higher binding energy also present in Supplementary Fig. 6 indicates functionalized and oxidized carbon species (C-O, C=O). **c** X-ray photoemission electron microscopy image normalized by the background intensity and collected at room temperature (top) using the Pd 3$d$ photoemission line at a photon energy of 410 eV (bottom) on LCC Pd with carbon sublayer. **d** Catalytic conversion and (**e**) selectivity of LCC model system Pd with and without carbon sublayer. **f** Representative layered structure of the studied system with/without carbon interlayer deposited on Si(100). Reaction condition: $C_2H_2$: 0.9 ml/min; $H_2$: 27 ml/min; $N_2$: 5.8 ml/min; T = 150 °C. Source Data.

This is a pre-requisite for the intended steady-state between carbon and activated hydrogen defining the catalyst's limits of operation in temperature, hydrogen to acetylene ratio, pressure and flow rate such as to maintain an optimal concentration of subsurface carbon. We note that the existence of carbon atoms on the reactive interface requires the existence of fragmented reactant molecules that as radicals must be prevented to form oligomers or carbon. Since such reactions are likely to occur on planar surface facets of Pd, it is important to design the Pd reactive interface as rough nanostructures[14,16] with small Pd ensembles containing a large number of edge and corner atoms being crucial to avoid oligomerization side reactions of acetylene.

To experimentally verify the role of carbon for the catalytic performance (Supplementary Note 3), two Pd films with and without contact to carbon were prepared followed by in-system thermal and X-ray photoelectron spectroscopy (XPS) studies (Samples S1, S2 and S3 in the sample table of Supplementary Information). Both samples are based on Si(100) coated by ~20 nm thermal $SiO_2$ followed by ~3 nm Pd deposited by evaporation (Supplementary Note 4). One sample was made with a ~0.4 nm carbon layer between $SiO_2$ and Pd. XPS C 1$s$ lines (Fig. 3b) indeed corroborated the absence of carbon for the C-free sample (lower spectrum). The C 1$s$ signal for the C-containing sample was deconvoluted into four contributions at 283.8 eV, 284.3 eV,

285.4 eV and 287.0 eV. The lowest binding energy species (283.8 eV) is typical for isolated carbon atoms dissolved in Pd[26]. The presence of carbon in the as prepared Pd LCC sample was also confirmed with transmission electron microscopy (TEM) lamella cross section studies (Supplementary Fig. 7g, f). In Supplementary Fig. 6a we report the spectrum of the carbon film without Pd exhibiting at 284.4 eV the signal for $sp^2$ carbon. The spectrum in Supplementary Fig. 6b (bottom) originates from the film used for Fig. 3b (also Supplementary Fig. 6b-top) but after thermal annealing to drive the carbon into the Pd, leading to an increase of the C 1$s$ signal at 283.8 eV, which was assigned to C incorporated into Pd. Such behavior is in line with the theoretical predictions (Fig. 3a and Supplementary Fig. 1). Figure 3c shows an X-ray photoemission electron microscopy (XPEEM) image of the surface recorded at the Pd 3$d_{5/2}$ energy showing the LCC motif with a granular Pd structure forming a coating on the (dark) support (also Supplementary Fig. 7b).

Both samples (Sample S1 and S3) were tested for acetylene hydrogenation at 150 °C. The sample without carbon shows a low initial conversion of 22% quickly dropping to almost zero (Fig. 3d, red dots). For the C-containing sample conversion started at 97% and dropped to 53% at 20 h time on stream (TOS) (Fig. 3d, blue dots). The selectivity to ethylene (Fig. 3e) increases with time, reaching over 90% $C_2H_4$ after 20 h. Full hydrogenation reaction dominates ($C_2H_6$ 73.4%) in

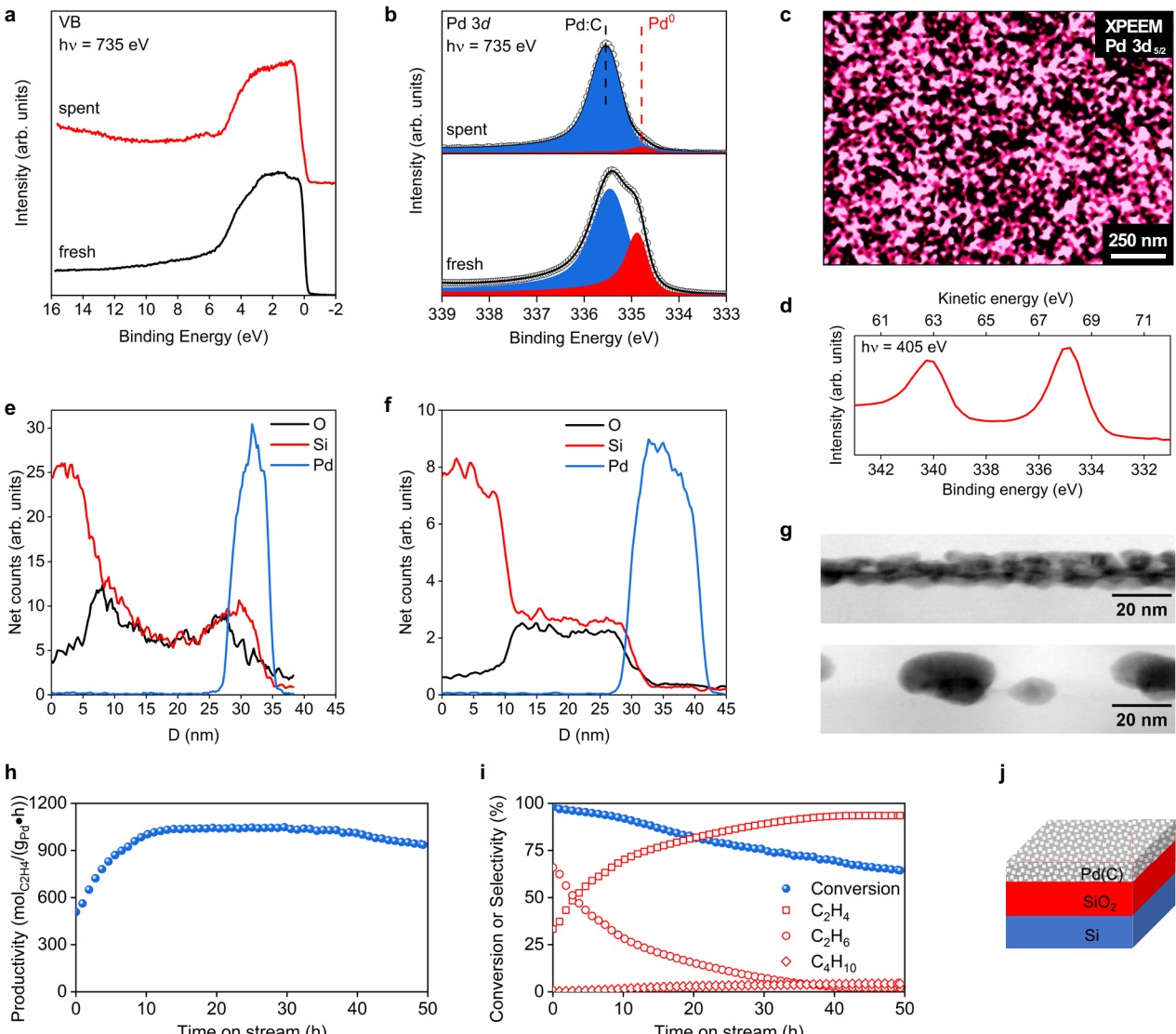

**Fig. 4 | Realization of the LCC catalyst. a** XPS valence band and (**b**) Pd $3d_{5/2}$ spectra of a Pd (3 nm) thin film deposited on $SiO_2$(20 nm)/Si(100) measured at RT under UHV (bottom) and after operando $C_2H_2$ hydrogenation at 1 mbar (top). The kinetic energy of the photoelectrons is 400 eV. **c** X-ray photoemission electron microscopy image normalized by the background intensity and collected at room temperature using the Pd $3d$ photoemission line. **d** Pd $3d$ photoemission spectrum collected at a photon energy of 405 eV. Cross-section EDS line scans (elements: Si, Pd, O) of (**e**) the as-prepared sample and (**f**) the spent sample (for more details see Supplementary Fig. 20). **g** Cross-section view tilted by 15° on both, fresh (top) and spent (bottom) Pd films, showing a dense fresh Pd film which transformed into sintered large NPs with less surface coverage in the spent film. **h** Productivity and (**i**) conversion & selectivity of the same catalyst (three data points are skipped between each dot for clarity). **j** Representative layered structure of the studied system with adventitious carbon incorporated into the Pd film during sample preparation. Reaction condition: $C_2H_2$: 0.9 ml/min; $H_2$: 27 ml/min; $N_2$: 5.8 ml/min; T = 150 °C. Source Data.

the first 2 h and drastically drops thereafter. The deactivation of the carbon containing sample is traced back to surface blocking by hydrocarbons (Supplementary Figs. 23, 28), since poly-acetylene was found in many analyses of carbon-modified films in this study. Post-reaction analysis revealed that the Pd film without carbon dissolved Si to convert into $Pd_3Si$ (Supplementary Fig. 9, Si $2p$ 98.52 eV, Pd $3d$ 336.63 eV). This phase is non-reactive in hydrogenation under the chosen reaction conditions. The dissolution of carbon from the functional interface into Pd prevented its hydrogenation and did allow sustained and selective semi-hydrogenation. However, carbon dissolution was insufficient to modify the whole Pd present, leaving some unselective activity, producing polymers that poison the catalyst. The model experiments confirm the validity of the design assumptions and highlight that both, the carbon concentration and the nature of the functional interface need attention in combination with the hydrogen

chemical potential of the reaction. In this study, we focus on the rational design of this kind of subsurface carbon in a well-controlled manner by both, an UHV preparation of a very thin carbon sublayer, as well as by sputtering Pd LCC in a carbon-rich (adventitious) environment. The subsurface carbon (Pd:C) in both samples is self-repairing during acetylene hydrogenation, which leads to an excellent catalytic performance (conversion, selectivity and productivity) comparable to that of powder catalyst but using much less catalyst material.

## Realization

A freshly synthesized 20 nm $SiO_2$ functional interface by Plasma-Enhanced Chemical Vapor Deposition (PECVD) prevented silicide formation with reactive Pd nanostructures incorporated with C during deposition (Supplementary Note 4). The thickness of Pd was fixed to 3 nm (Fig. 4e, Supplementary Fig. 20b). Figure 4 summarizes the

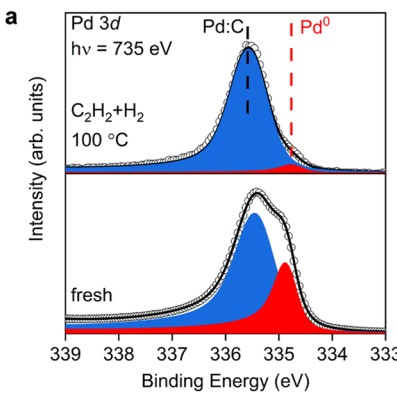

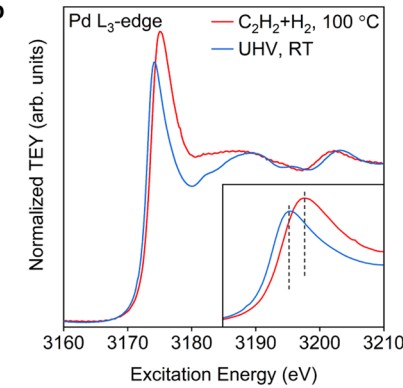

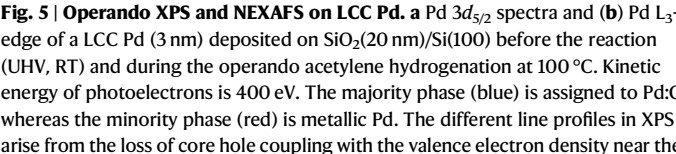

**Fig. 5 | Operando XPS and NEXAFS on LCC Pd. a** Pd $3d_{5/2}$ spectra and (**b**) Pd $L_3$-edge of a LCC Pd (3 nm) deposited on $SiO_2(20$ nm)/Si(100) before the reaction (UHV, RT) and during the operando acetylene hydrogenation at 100 °C. Kinetic energy of photoelectrons is 400 eV. The majority phase (blue) is assigned to Pd:C whereas the minority phase (red) is metallic Pd. The different line profiles in XPS arise from the loss of core hole coupling with the valence electron density near the Fermi edge as a consequence of the Pd:C formation (see Fig. 4a). The broad line for Pd:C in the NEXAFS indicates the inhomogeneity of the distribution of carbon giving rise to locally different electronic structures (see also theory in Fig. 3a and Supplementary Fig. 1). Note that the depth of information is strongly different for XPS and NEXAFS; XPS at 400 eV kinetic energy probes surface-sensitive, whereas the NEXAFS measures through the whole Pd nanostructures. Source Data.

characterization results (Supplementary Notes 5–7) of a typical LCC catalyst reproduced over 100 times throughout this study (Sample S4). In the valence band spectra (Fig. 4a) the Fermi edge of Pd is well developed in the fresh nanostructure but is greatly reduced in intensity after reaction. The corresponding shift of electron density is characteristic of C dissolved into the metal[34] and also causes a pronounced change in core level line shapes (Fig. 4b). The background and the feature below 6 eV arise from an overlayer of carbonaceous species. This state of a reacting Pd surface was often found before[34] and gave rise to the idea of the "flexible surface" promoted by G.A Somorjai[31], in which the carbonaceous fragments are not only spectators but may control the selectivity[35] through a modified adsorption of the reagents. In the Pd $3d_{5/2}$ spectra (Fig. 4b) two contributions are found, metallic Pd at either 334.9 eV for fresh or 334.7 eV for used samples, and a signal for Pd:C at 335.5 eV[34] (Supplementary Note 5). The contribution of Pd:C increased during the reaction.

Figure 4c shows an XPEEM image recorded at the Pd $3d_{5/2}$ energy revealing a mesoscopically flat and dense film. At the nanoscale, however the film exhibits a granular structure forming a carpet on the (dark) support (Supplementary Figs. 20a, 22). This morphology reveals the intended strong interaction at the functional interface and was also found in the model systems (Fig. 3c) made by evaporation. Supplementary Fig. 20b displays high-resolution transmission electron microscopy (HRTEM) cross-sectional views of the fresh homogeneous film with average Pd thickness of 3 nm. The presence of polycrystalline cubic Pd is evidenced by the ring Fast Fourier transform (FFT) diffraction pattern (Supplementary Fig. 20b). The granular structure seen in the XPEEM and SEM manifests itself in the variable contrast in the high-angle annular dark field (HAADF) image of Supplementary Fig. 20b. By tilting the lamella by 15 deg, the granular isotropic structure of the LCC becomes visible (Fig. 4g, top, and Supplementary Fig. 20b), resulting in a rough, stepped surface. This is in accord with our IR data unveiling the presence of low-coordinated surface sites that favor the binding of CO in bridge-mode configuration (Supplementary Fig. 29). Figure 4g (bottom) reveals the formation of some Pd islands for the spent LCC, many of which retain their functional interface with the $SiO_2$ supported by energy dispersive X-ray spectroscopy (EDS) in Fig. 4e, f. The EDS line scans display that Pd, Si and O partially overlap before catalysis indicating that Pd sits in $SiO_2$ bowls. After catalysis the overlap is almost absent, implying that the $SiO_2$ layer has flattened, while Pd forms segregated nanoparticles on top. This shows that the $SiO_2$ support unexpectedly restructures at the low

operation temperature and partly loses its function at the functional interface. Supplementary Fig. 20c confirms this for the spent LCC. The limited stability of some Pd nanostructures is ascribed to the found impurity of the LCC still containing metallic Pd besides Pd:C (Fig. 4b). Unmodified Pd is converted during activation to PdH that destroys the local functional interface by hydrogenation, ending up as 3-dimensional nanoparticles.

Sustained catalytic activity was achieved with these LCC films. Figure 4h reports the productivity and Fig. 4i the respective data for conversion and selectivity. At the start of the catalytic test, with conversion near 100%, the selectivity toward undesired ethane ($C_2H_6$) exceeds that of ethylene ($C_2H_4$). Over time, this trend reverses, with a more pronounced improvement for Pd LCC as compared to bulk materials (Supplementary Fig. 30). This catalyst formation phase involves atomic carbon entering octahedral sites in the Pd lattice promoting selective acetylene hydrogenation as predicted by DFT[36,37]. Productivity rises sharply during the catalyst formation phase from the beginning of operation to the 10th h. A steady phase is observed until 40 h TOS, where productivity is about 1040 $mol_{C2H4}/(g_{Pd}\cdot h)$. Later, productivity starts to decline to 937 $mol_{C2H4}/(g_{Pd}\cdot h)$ at 50th h further decreasing for one week (Supplementary Fig. 24). Comparison of the performance of LCC catalysts with Pd closed thin films, a Pd foil and selected reference data in different metrices confirms the design success of the LCC (Supplementary Fig. 25). The formation of carbonaceous deposits is not the only reason for deactivation, while the activity for saturated hydrocarbon formation of the LCC film is another reason, as evidenced by the carbon balance and the thermal analysis (Supplementary Fig. 23), although being lower than that of dispersed Pd/$SiO_2$ (Supplementary Fig. 24g). The accumulation of carbon deposits on the sample surface after extended operation time (260 h) was revealed by Raman spectroscopy (Supplementary Fig. 23).

The improvement arises from adapting the activation gas composition such as to maximise the Pd:C content (Supplementary Fig. 26). Strong support comes from an operando experiment in 1 mbar where a standard LCC was investigated during acetylene hydrogenation. Figure 5a reports the Pd:C phase as active phase as its abundance grows during operation in agreement to similar observations[38] at higher pressure (for C 1s, O 1s, Pd 3p, Si 2p, see Supplementary Fig. 17). The increase in Pd:C is also validated by operando NEXAFS measurements (Fig. 5b). The fresh sample has spectral features assigned to metallic Pd with a slightly shifted and broader whiteline compared to that of the Pd foil, due to differences in

**Table 1 | Performance of LCC Pd and comparison to bulk catalysts**

| Entry | Sample | Treatment | Productivity $C_2H_4$ (mol/$g_{Pd}$/h) | Productivity $C_2H_4$ (mol/m²/h) | Ratio-Semi/full hydrogenation rate* |
|---|---|---|---|---|---|
| 1 | S4 LCC Pd | None | 1059 | 28.8 | 4.8 |
| 2 | S4 LCC Pd | $SiO_2$ 473 K, UHV | 1045 | 28.4 | 24.0 |
| 3 | S4 LCC Pd | Pd 473 K, UHV | 1002 | 27.3 | 4.9 |
| 4 | S4 LCC Pd | $SiO_2$ 473 K, UHV Pd 473 K, UHV | 929 | 25.3 | 49.4 |
| 5 | S4 LCC Pd | $SiO_2$ Ar sputt. | 695 | 18.9 | 14.1 |
| 6 | S5 100 nm Pd | None | 20 | 23.6 | 1.3 |
| 7 | S6 foil Pd | None | 0.015 | 15.2 | 2.5 |
| 8 | S7 powder Pd | None | 1044 | - | 2.6 |

Productivity of LCC Pd and reference materials (grey rows) studied in the present work. Productivities for $C_2H_4$ and $C_2H_6$ were measured at 18 h of time-on-stream. Synthesised material system: $Si(100)/SiO_2(20\ nm)/Pd(3\ nm)$: 3 mm × 20 mm, $Si(100)/SiO_2(20\ nm)/Pd(100\ nm)$: 3 mm × 20 mm, Pd foil: 3 mm × 20 mm, Pd powder: 0.016% Pd@$SiO_2$, 9 mg. Reaction condition: $C_2H_2$: 0.9 ml/min; $H_2$: 27 ml/min; $N_2$: 5.8 ml/min; T = 150 °C. (* formation rate of $C_2H_4$ divided by formation rate of $C_2H_6$ (i.e. partial versus total hydrogenation)). The detailed information on samples is in Supplementary Information.

**Fig. 6 | Robustness of catalyst selectivity.** Shown is a design-of-experiment assessment of various factors on the stability of the LCC catalysts after an additional heat treatment (Supplementary Note 9). Summarized is the effect of 10 individual factors and the identified two most influential pairwise interactions in the initial preparation and subsequent conditioning.

the particle size and carbon content (Supplementary Fig. 14b)[17]. During the reaction at 100 °C, the whiteline shifts to higher energies and broadens which is a distinct signature of Pd:C phase under alkyne hydrogenation conditions[39].

The stability of the functional interface can also be optimized by a pre-reduction at mild conditions, whereby overheating leads to a reduction of the oxide support and a strong loss in hydrogenation performance (Supplementary Fig. 27). To study chemisorption at the LCC, we performed experiments with CO and acetylene at 310 K (Supplementary Note 8) and investigated the adsorption of CO with polarization modulation infrared reflection adsorption spectroscopy (PM-IRAS) (Supplementary Fig. 29). Using CO as probe molecule for the Pd state after the standard hydrogen activation, the activated surface appears rough and stepped, which is in agreement with TEM (Fig. 4g, top, and Supplementary Fig. 20b), and far from the observations previously made for GaPd systems[20] that are characterized by a facetted state. Spectra of adsorbed CO in Supplementary Fig. 29 reveals for LCC only bridge CO adsorption sites (bands between 1960 and 1970 cm⁻¹) as observed for higher indexed Pd(210) surfaces[40] or defect-rich Pd nanoparticles[41]. In contrast, a mixture of hollow (1822 cm⁻¹), bridge (1928 cm⁻¹) and on-top sites (2054 cm⁻¹) is detectable on Pd foil (Sample S6), typical for Pd(111) surfaces[40]. The peak shift for bridge-bonded CO to higher wavenumbers for the LCC catalyst compared to the Pd foil suggests weaker back-bonding and thus a less electron-rich Pd surface. The higher metallic character of the Pd foil in comparison to the LCC catalyst is also confirmed via valence band spectra, showing higher density of states at the Fermi edge for the Pd foil (Fig. 4a, and Supplementary Fig. 19). A rough irregular structure in the LCC catalyst is in line with its design concept to avoid extended sites for hydrogen dissolution and for oligomer formation.

## Performance

Application in carbon management processes requires fast flows of large volumes and thus, highly active and selective catalysts. The planar non-porous nature of our LCCs qualifies for reactors working with turbulent gas flows. Preliminary kinetic data taken at low conversion reveal that the performance is limited by mass transport phenomena. More suitable reactors and removal of the phase impurity of elemental Pd are pre-requisites for proper kinetic analysis. Thus, we confine the report here in Table 1 to performance data (Sample S4) and a comparison with other catalysts (Supplementary Note 7) prepared in our laboratory (Samples S5, S6, S7) and suggested for this reaction (catalysis data in Supplementary Fig. 30a–c). Table 1 summarizes detailed results also available in the corresponding Supplementary Tables 7 and 8.

A closed sputter-deposited Pd film (entry 6) is a poor catalyst, but already better than a Pd foil (entry 7). High dispersion and fixation of Pd nucleation structures either chemically (entry 8) or by sputtering (entry 1) creates highly productive systems but with low selectivity. Only if the functional interface is modified from the native[42] hydroxylated SiO$_2$ state (entries 2,4,5), truly productive systems are obtained, allowing the existence of the Pd:C phase. Sputtering the SiO$_2$ film (entry 5) or annealing after deposition (entry 3) modifies the performance marginally. Combining annealing of the fresh SiO$_2$ plus the deposited Pd, both under UHV, results in best selectivity. With a comparable selectivity, our productivity is outstanding and at least an order of magnitude better than many benchmark powder systems, as indicated in Supplementary Table 1 for similar reaction conditions. Analysis of semi/full hydrogenation rate vs. acetylene consumption rate in Supplementary Fig. 30c reveals the superior performance of LCC to conventional powder, bulk Pd as well as Pd foil catalysts, and highlights the potential of designing the functional interface in LCC geometry.

## Optimization strategy

On the basis of the described results, we intend to develop a strategy for further rational optimization of the LCC Pd catalyst stability. Therefore, we employ efficient, robust design-of-experiment (DoE) effect analysis (Supplementary Note 9) to assess the key factors determining the catalytic performance of the Pd nanostructures with minimum number of experiments. Specifically, we hereby consider the parameters that control the deposition of the Si(O$_x$) buffer layer and the Pd layers (total pressure applied, plasma or sputtering power) and factors governing the initial conditioning of the catalysts under different reaction conditions (presence or absence of carbon sources) as well as different temperature profiles (i.e. Samples S8). The difference in selectivity between semi- and full-hydrogenation before and after the conditioning protocol is used as stability descriptor. The effect analysis in Fig. 6 reveals the sputter power during Pd deposition and the presence of acetylene and hydrogen during the formation phase as key design factors that decisively affect selectivity and thus stability of the final catalyst. Most intriguingly, however, our analysis also reveals an entanglement of these factors exhibiting significant pairwise interactions. Suitably matching up the acetylene feed during the conditioning with the sputter power during the deposition allows for instance to directly improve the stability of the LCC catalyst, instead of the otherwise negative effect of this factor. Deliberately employing the conditioning as an active part of the catalyst design process offers therefore hitherto unchartered possibilities for a further optimization, which we now aim to systematically explore.

In summary, our study unveils that LCC catalysts with a high 2D density of metastable Pd nanostructures, exhibiting a strong functional interface and the ability to maintain an optimum level of dissolved carbon, can provide the functionality required for the selective hydrogenation of concentrated acetylene streams. Carbon can retain high rates of semi-hydrogenation[30] of bare Pd. Its dynamic instability in the sub-surface regime can be "self-repaired" by the reagents. We verified that a large geometric surface area is not necessary for interfacial catalysts if a high density of active sites can be maintained. This unexpectedly requires the lateral separation of nanostructures realized in the LCC approach.

The excellent performance achieved so far prompted us to devise a data-driven approach for developing future LCC based on realistic metal supports and produced in such large dimensions that demonstration reactors with adapted flow characteristics can be realized. With this, future carbon management systems can be designed circumventing the carbon capture and storage (CCS) approach. Solid carbon can safely be stored, be used for soil improvements or in building materials, whereas the platform molecules methanol and ethylene can support future chemical industries

or serve as e-fuels and so pay back at least some of the cost for energy necessary to split the oxygen from the carbon in the CO$_2$ to be stored. The LCC catalysis concept further provides a unique opportunity to learn about the usually hidden support interactions in catalysis. The effect of tailoring the sub-surface volume and of controlling sub-surface species will provide a new dimension of catalyst control through auto-tuning of (metal) catalysts by dissolving reactant atoms. The excellent performance observed with LCC systems would allow to move this technology to the area of microchannel systems and building "on-chip" sensors for online reaction monitoring. For catalysis science we expect deeper mechanistic insight of the reaction by removing the material gap for operando analysis, a historic caveat[7], when using LCC as a broader concept applicable to many (metal) catalyst systems.

## Methods

### DFT calculation details

All DFT calculations are performed with the Quantum Espresso[43,44] (version 6.6) software package and the revised Perdew-Burke-Ernzerhof (RPBE) exchange-correlation functional[45]. A plane wave basis set describes the electronic states with an energy cutoff value of 680 eV for both Pd-fcc bulk and all surface structures. Brillouin zone sampling uses Fermi-Dirac smearing along with $12 \times 12 \times 12$ and $4 \times 4 \times 1$ Monkhorst-Pack grids[46] for bulk and surface slab structures, respectively.

Using the described computational setup, the optimized lattice constant of bulk fcc Pd is 3.970 Å. Slabs were then constructed in the (111) and (100) facet orientations as $3 \times 3$ supercells and 4 layers in thickness. The bottom two layers were constrained and the top two were allowed to relax within convergence criteria $\leq 0.01$ eV/Å for atomic forces and $\leq 0.001$ Å for the total energy using the Broyden–Fletcher–Goldfarb–Shanno (BFGS) algorithm. A vacuum region of at least 16 Å is included between slabs in the non-periodic $z$-direction, along with a dipole correction to compensate for artificial adsorbate-induced electrostatic interactions[47].

### Model system: thin film synthesis (evaporation) and measurements

All sample preparation steps have been performed under ultra-high vacuum (UHV) conditions to avoid contaminations. The substrates used here were Si(100) wafers with a size of $10 \times 10$ mm². Prior to introduction into the UHV chamber, the wafers were pre-cleaned during 20 min in an ultrasonic cleaner with acetone at 45 °C, followed by 6 min of soaking in methanol. After wet cleaning, the samples were loaded into the UHV chamber.

Here, the samples underwent three sputtering annealing cycles, consisting each of sputtering with 3 keV Ar ions for one hour followed by annealing at 550 °C for 20 min. The SiO$_2$ layer was prepared by thermal oxidation in an integrated high-pressure cell (HPC). In the HPC, the wafers were heated for three hours at 690 °C in an O$_2$ ambient atmosphere of 650 mbar. In this process, the temperature was raised gradually from room temperature to 690 °C within 30 min. The resulting SiO$_2$ layer was ~20 nm. Carbon was deposited in the UHV chamber with an electron beam evaporator employing a 1 mm carbon rod. The sample was maintained at room temperature at a chamber pressure of ~$2 \times 10^{-8}$ mbar. XPS was employed to calibrate the C layer thickness. Pd was also deposited with an electron beam evaporator in the UHV chamber. The sample was kept at room temperature, and the base pressure of the chamber was $4.4 \times 10^{-9}$ mbar. The Pd deposition rate was calibrated with a Quartz microbalance.

The as-deposited films were checked with respect to thickness and composition with XPS employing monochromatized Al$_{k\alpha}$ radiation and a Specs electron energy analyzer (Phoibos 200). For quantitative evaluation, the spectra were fitted using the CasaXPS software[48].

The UHV-prepared $Pd/C/SiO_2$ samples were moved via glovebox transfer to a quartz-bed plug flow reactor and tested for acetylene hydrogenation at 150 °C. This inert sample transfer was performed in order to prevent the modification of the well-defined carbon content purposely deposited in these samples.

## Preparation thin film synthesis (sputtering)

Additional thin film catalysts used in this work were synthesized using commercial relevant deposition technologies namely physical vapor deposition (PVD, sputtering) and plasma-enhanced chemical vapor deposition (PECVD). Czochralsky grown, 6- and 8-inch silicon wafers were used as substrate material.

To separate the wafer into individual sample pieces, we employed UV laser ablation (Keyence MD-U) to structure the backside of the wafers. After structuring, the samples were separated into 5×5 cm² plates and wet chemically cleaned afterwards. For cleaning a conventional RCA (Radio Corporation of America) treatment was performed by submerging the samples in an ammonia solution, followed by aqueous hydrogen peroxide solution. Both steps were performed at 75 °C for 7 min. To remove the native oxide formed during the steps, an HF dip (1% HF) and DI water rinse was performed. Final native oxide was removed immediately before coating of the silicon dioxide. Silicon dioxide buffer was deposited by PECVD using 60 MHz excitation at 200 W. Precursor gases were monosilane ($SiH_4$) and nitrous oxide ($N_2O$) at flow ratio of 1:40. Substrate temperature was kept constant at 400 °C.

After buffer deposition, samples were transferred under $N_2$ atmosphere into a magnetron sputter coater (PREVAC) for palladium deposition at RT. Nominal 3 nm (with accuracy of ±1 nm) thin films were deposited using a 5 N 2-in palladium target, Argon flow rate of 1.8 sccm, chamber pressure of $4 \times 10^{-3}$ mbar and a 13.56 MHz RF plasma operated at 30 Watts.

## Synthesis of Pd/SiO₂ powder reference catalysts

Reference $Pd/SiO_2$ catalysts were synthesized by incipient wetness impregnation of aqueous solutions containing the corresponding amount of Pd(II) nitrate dihydrate (40% Pd basis, Merck) on $SiO_2$ (Davisil grade, pore size: 60 Å, particle size: 35–60 mesh; Sigma-Aldrich). $SiO_2$ support was calcined at 550 °C for 6 h prior to impregnation. The resulting mixtures were subsequently dried at 80 °C for 20 h and heat-treated at 560 °C for 4 h in air flow (100 mL/min).

## XPS

X-ray photoelectron spectroscopy and X-ray absorption spectroscopy measurements were performed at the BElChem (UE56/2-PGM1)[49] and CAT at EMIL[50] (UE48-PGM and U17-DCM) endstations of BESSY II, Berlin. The APXPS setup equipped with a differential pumping station and SPECS PHOIBOS 150 NAP hemispherical analyzer allows to perform operando measurements in different atmospheres. More details of the experimental setup are provided elsewhere[51]. When required, the sample is heated with an IR laser from the backside and the temperature is measured with a thermocouple placed on the sample. Binding energies are calibrated with respect to the Fermi edge measured at the same photon energy as the corresponding core level.

The Pd $3d$ spectra are fitted with Doniach-Sunjic functions with different asymmetry parameters for each component. For deconvolution of peaks, CasaXPS[48] and a custom-made fitting software using the non-linear least squares fitter from the lmfit package[52] written in Python was used. For the metallic Pd component, an asymmetry parameter calculated by Hüfner et al.[53] was used. For the Pd:C peak, a lower asymmetry parameter was used to be able to fit the peak. Pd $3p$ peaks are also fitted with Doniach-Sunjic function while O $1s$ peaks are

fitted with mixed Gaussian-Lorentzian line shapes in 1:1 ratio. The C $1s$ spectra are fitted using Doniach-Sunjic functions with different asymmetry values, based on the literature on carbonaceous species[54,55]. For the operando XPS/XAS measurements during selective acetylene hydrogenation reaction, 1 mbar $C_2H_2 + H_2$ is dosed in the analysis chamber by back-filling method with a ratio of 1:10. The energy alignment during the operando measurements is performed based on the first Fermi edge observed in the valence band on lower binding energy side. XAS measurements are performed in the TEY mode. Background of Pd $L_3$-edge spectra is normalized by Athena software (Demeter 0.9.26 package)[56]. The probing depth of measurement is several nm.

## X-ray photoemission electron microscopy (XPEEM) and low energy electron microscopy (LEEM)

The experiments were carried out in the SMART microscope operating at the UE49-PGM beamline of the synchrotron light source BESSY II of the Helmholtz Centre Berlin (HZB). The aberration corrected and energy filtered LEEM/PEEM instrument combines microscopy (LEEM/XPEEM), diffraction (μ-LEED), and spectroscopy (μ-XPS) techniques for comprehensive characterization. The base pressure of the system is $10^{-10}$ mbar, but operation is possible at pressures up to $10^{-5}$ mbar of reactive gases in a temperature range between 150 and 1500 K[57–59].

Once the samples were synthesized, they were transferred to the SMART microscope in inert Ar atmosphere, in order to keep the carbon and oxygen contamination to a minimum. For the same reason, sample transfer between different sample holders was performed in an Ar glove box. Samples were then introduced in a load lock chamber at the SMART microscope with the help of an Argon filled glove bag that was purged multiple times prior to sample manipulation. After that, the sample was transferred to the analysis chamber where the base pressure was kept at $2 \times 10^{-10}$ mbar during experiments.

## Volumetric adsorption

Adsorption energies of CO were determined at 40 °C using a BT2.15 Tian-Calvet type Calorimeters (SETARAM, E14). The calorimeters are equipped with a custom-designed high vacuum and gas dosing apparatus (volumetric-barometric system) and an in-house constructed high vacuum all-metal cell with batch geometry[60,61].

This configuration allowed probe molecules dosages as small as 0.02 μmol. A pressure transducer (MKS Baratron type 121) was used to detect pressure variations of 0.001 mbar. The calorimetric sensor was calibrated using a contact-free method based on the Joule effect. A dedicated vessel with a built-in electrical heater (Ohmic resistance 1kΩ) was used to simulate the experimental vessel containing the sample, and calibration was carried out using a series of precise heat inputs. The LCC catalysts were cut to a size of 1 × 5 cm to ensure uniform measurement conditions.

First, the LCC catalysts were reduced in the calorimeter cell at 125 °C in 300 mbar and 30% $H_2$ for 30 min following by evacuation at 125 °C for 30 min and cooling down to 40 °C ($p < 3 \times 10^{-8}$ mbar). CO was stepwise introduced into the evacuated cell ($p < 3 \times 10^{-8}$ mbar), and the pressure evolution and the heat signal were recorded for each dosing step.

The adsorption run is finished when the calculated heat of adsorption approaches the enthalpy of condensation of the probe molecule, and the associated adsorption isotherm shows a plateau (all adsorption sites are occupied). The first adsorption run is followed by a desorption process performed by evacuation of the calorimetric cell ($p_{final}$ ~ $10^{-8}$ mbar). In this way, the reversible adsorbed phase desorbs and either the pristine surface is restored, in case of an entirely reversible adsorption or the pristine surface is not recovered, in case of

a (partially) irreversible adsorption. Next, the 2nd adsorption run is performed to assess which fraction (if any) of the pristine surface sites are irreversible occupied by the adsorbed phase. The stoichiometry was determined by means of corresponding IR spectroscopic experiments.

## PM-IRAS

Polarisation-modulation infrared reflection absorption spectroscopy (PM-IRAS) was performed using a VERTEX80v FTIR spectrometer system (Bruker Optik GmbH) coupled with an external polarization-modulated application unit (PMA50) and a RefractorReactor™ Grazing Angle Accessory from HARRICK for the adsorption of probe molecules and for operando studies of surface reactions at elevated temperatures and normal pressure. The LCCs with a dimension of $25 \times 30$ mm was stored in a desiccator under vacuum until investigation. The sample was placed into the cell, aligned by maximizing the detector signal, and pretreated at 125 °C (heating rate 5 °C/min) in presence of an equilibrium pressure of 300 mbar $H_2$ for 1 h followed by evacuation at 125 °C for 30 min and cooling to 40 °C. CO was dosed stepwise into the cell up to 300 mbar equilibrium pressure followed by desorption at 40 °C to a residual pressure of $3 \times 10^{-6}$ hPa. Spectra were recorded at different equilibrium pressures and after removal of the gas phase and desorption of weakly bonded CO by evacuation at 40 °C. PM-IRAS data were acquired at 10 kHz interferometer velocity, accumulating 2048 scans in double-sided, forward-backward mode at a resolution of 4 cm$^{-1}$ at an angle of 75° from the surface normal using a MCT IR detector D316/B. The PEM controller setting (peak retardation) was 1700 cm$^{-1}$ to optimize efficiency of the photo-elastic modulation for the spectral region of interest. Demodulation of the sum and difference spectral channels was carried out by using the PM-IRAS software from Bruker applying the baseline correction method.

## Catalysis

Activity of the LCCs was tested using acetylene hydrogenation as the probe reaction. Only the UHV-deposited model Pd LCC samples with a defined amount of carbon were transferred to the reactor using an inert transfer method. The remaining samples were transferred in air, after confirming that this transfer did not affect their characteristics. The acetylene hydrogenation reaction was performed in a quartz-bed plug flow reactor with LCC catalyst (60 mm$^2$) vertically sitting in the reactor. The inner diameter of the reactor is 7 mm. Quartz wool was put on the top and bottom of LCC catalyst. The as-synthesized catalysts were pretreated in 30% $H_2$ ($N_2$ balance, flow rate 30 mL/min) from room temperature to 125 °C at a heating rate 5 °C/min and dwell for half-hour. The pretreatments on LCC in Ar, $C_2H_2$, $C_2H_4$ and at 125, 150, 200, 300 °C were compared. Later, temperature was cooled/heated in $N_2$ atmosphere to the target temperature for catalysis. Catalytic reactions were performed isothermally at temperatures of both 150 and 80 °C and a gas mixture of 2.67% $C_2H_2$, 80% $H_2$ balanced with $N_2$. The total flow of the gas mixture was 33.7 ml/min. LCC catalysts were tested from at least 18 h to long-time stability two weeks. The exhaust gas after catalysis was analyzed by an online Agilent gas chromatograph equipped with a capillary column and a flame ionization detector (FID) (Agilent 7890). The following GC column combinations were used for product analysis: Plot-Q (length 30 m, 0.53 mm internal diameter, 40 μm film thickness) plus FFAP (length 30 m, 0.53 mm internal diameter, 1 μm film thickness) connected to a flame ionization detector (FID) for analysis of hydrocarbons.

As a comparison, Pd foil with the same size and Pd/SiO$_2$ powder catalysts were tested in the same reactor and catalytic conditions. The catalyst bed of Pd/SiO$_2$ powder was kept to the same height (2 cm) and with Pd amount (1.26 μg) very close to that in LCC (1.63 μg).

Conversion and selectivity were calculated as following:

$$X = \frac{C_2H_2(in) - C_2H_2(out)}{C_2H_2(in)} \times 100\% \tag{1}$$

$$S_{C2H4} = \frac{C_2H_4(out)}{C_2H_4(out) + C_2H_6(out) + 2 \times C_4H_{10}(out)} \times 100\% \tag{2}$$

$$S_{C2H6} = \frac{C_2H_6(out)}{C_2H_4(out) + C_2H_6(out) + 2 \times C_4H_{10}(out)} \times 100\% \tag{3}$$

$$S_{C4H10} = \frac{2 \times C_4H_{10}(out)}{C_2H_4(out) + C_2H_6(out) + 2 \times C_4H_{10}(out)} \times 100\% \tag{4}$$

where $C_xH_y$ are the gas concentrations analyzed by gas chromatograph before or after reaction, respectively. $X$ and $S$ are conversion and selectivity, respectively.

Carbon balance ($B$) is calculated by the comparison between the carbon in feed (here $C_2H_2$) and carbon in $C_2H_4$, $C_2H_6$, $C_4H_{10}$ and unconverted $C_2H_2$ in product gas.

$$B = \frac{C_2H_4(out) + C_2H_6(out) + 2 \times C_4H_{10}(out) + C_2H_2(out)}{C_2H_2(in)} \tag{5}$$

The $C_2H_4$ productivity of the catalysts was calculated as follows (either normalization by mass of Pd or area of Pd):

$$P_{mass} = \frac{(f/V_m) \cdot C_{C_2H_2} \cdot X \cdot S_{C2H4}}{m_{Pd}} \tag{6}$$

$$P_{area} = \frac{(f/V_m) \cdot C_{C_2H_2} \cdot X \cdot S_{C2H4}}{A_{Pd}}, \tag{7}$$

where $f$ is the flow rate of the feed; $V_m$ is molar volume of the ideal gas; $C$ is concentration of acetylene in the feed; $A_{Pd}$ is the area of thin film coated with Pd; $m_{Pd}$ is the mass of Pd in the catalyst which was measured by ICP-OES for both LCC Pd and Pd powder catalyst. $m_{Pd}$ is calculated as the mass of catalyst for Pd foil.

The number of the accessible Pd centers ($N$) was measured by CO adsorption. Here the corresponding acetylene consumption rate ($R$) was used to calculate turn over frequency (TOF).

$$R_{C2H2} = \left(\frac{f}{V_m}\right) \cdot C_{C_2H_2} \cdot X \tag{8}$$

$$TOF = \frac{R_{C2H2}}{N_{Pd}} \tag{9}$$

## TEM

For transmission electron microscopy (TEM) studies, an FEI Talos F200X instrument was used. This microscope is operated at 200 kV acceleration voltage and equipped with a SuperX Energy Dispersive Spectrometry (EDX) system incorporating four Silicon Drift Detectors (SDDs) detectors. High resolution transmission electron microscopy (HRTEM) imaging was accompanied by Fast Fourier transform (FFT) patterns in order to extract local structural information from the samples. In addition, scanning transmission electron microscopy (STEM) was employed in combination with energy dispersive X-ray spectroscopy (EDS) analysis in order to study the elemental composition of the systems. For all samples, EDS line scans and elemental maps of various regions were acquired. The FEI Ceta camera was used for HRTEM imaging, while the high-angle annular dark field (HAADF)

detector with a camera length of 98 cm was used for STEM imaging. HRTEM images were also taken on a double aberration-corrected JEOL JEM-ARM200F TEM equipped with a high-angle silicon EDS detector. The samples were prepared as lamellae for cross-sectional TEM by focused ion beam/scanning electron microscopy (FIB/SEM) using amorphous carbon as a protective layer on the top of the film.

## SEM
The SEM images were captured with a Hitachi s4800 applying a voltage of 1.5 kV at a working distance of 3 mm and using the secondary electron detector to visualize surface topography and morphology. The characteristic X-rays generated at an excitation energy of 10 kV (SEM-EDX analysis/QUANTAX 800 mit XFlash®6 (Bruker)) were used to determine the local elemental composition (EDS analysis).

## Data availability
The authors declare that the data supporting the findings of this study are available within the paper and its Supplementary Information files. Schematic diagrams are provided for Figs. 1, 2 and 6 in the paper. Source data for Figs. 3–5 are included with this paper. Additional data are available from the corresponding author upon reasonable request. Source data are provided with this paper.

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

## Acknowledgements

This project was funded by the German Federal Ministry of Education and Research (Bundesministerium für Bildung und Forschung, BMBF) under Grant No. 03EW0015B (CatLab). The authors appreciate the administrative coordinating role of Dr. Steffi Hlawenka for this project. We also acknowledge Zahra Gheisari for additional SEM characterization, Dr. Andrey Tarasov and Jasmin Allan for TG-MS measurements, and Dr. Thomas Götsch for developing the custom-made XPS fitting software. The Helmholtz-Zentrum Berlin is gratefully acknowledged for providing beamtimes at the synchrotron radiation facility BESSY II and the Energy Materials In-situ Laboratory Berlin (EMIL), jointly funded by HZB and the Max Planck Society, is acknowledged for sample preparation and (pre-)characterization.

## Author contributions

Data interpretation and measurements by Z.L. (catalysis), E.Ö., R.B., A. J., M.S., P.Z. (X-ray spectroscopy), A.E. (beam scientist), M.J.P., T.S. (XPEEM), C.R., M.D., A.H., W.F., F-P.S. (TEM/SEM), O.V.V., A.M.D. (Theory), C.K., M.G., P.K., F.G. (DoE), S.W., J.K., S.J. (IR, Volumetric Adsorption, Raman Spectroscopy); Sample design and preparation by A.S., A.J., M.M., D.D., T.K., M.A., M.B., J.F., T.M.K., R.G.-D.; Guidance and discussion of the results by K.S., V.J.B., D.A., C.S., T.L., H.K., A.K.-G, T.S., A.T., R. Schlatmann, K.R., B.R.C., R.S.; Manuscript preparation by K.S., Z.L., E.Ö., B.R.C., R. Schlögl with input of all co-authors. K.S., Z.L. and E.Ö. contributed equally.

## Funding

## Competing interests

The authors declare no competing interests.

## Additional information

Zehua Li [1,10], Eylül Öztuna [1,2,10], Katarzyna Skorupska [1,10] ✉, Olga V. Vinogradova[3], Afshan Jamshaid[4], Alexander Steigert[5], Christian Rohner [1], Maria Dimitrakopoulou[1], Mauricio J. Prieto[4], Christian Kunkel[3], Matus Stredansky[1], Pierre Kube[1], Michael Götte[5], Alexandra M. Dudzinski [3], Frank Girgsdies [1], Sabine Wrabetz[1], Wiebke Frandsen[4], Raoul Blume[1,6], Patrick Zeller[1,2], Martin Muske[5], Daniel Delgado [1], Shan Jiang[1], Franz-Philipp Schmidt [1], Tobias Köhler[5], Manuela Arztmann[5], Anna Efimenko[7], Johannes Frisch[7], Tathiana M. Kokumai[7], Raul Garcia-Diez [7], Marcus Bär [7,8,9], Adnan Hammud [1], Jutta Kröhnert[1], Annette Trunschke [1], Christoph Scheurer[3], Thomas Schmidt[4], Thomas Lunkenbein [1], Daniel Amkreutz[5], Helmut Kuhlenbeck [4], Vanessa J. Bukas [3], Axel Knop-Gericke[1,6], Rutger Schlatmann [5], Karsten Reuter [3], Beatriz Roldan Cuenya[4] & Robert Schlögl[1] ✉

[1]Department of Inorganic Chemistry, Fritz-Haber Institute of the Max Planck Society, Berlin, Germany. [2]Bessy II, Helmholtz-Zentrum Berlin für Materialien und Energie GmbH, Berlin, Germany. [3]Theory Department, Fritz-Haber Institute of the Max Planck Society, Berlin, Germany. [4]Department of Interface Science, Fritz-Haber Institute of the Max Planck Society, Berlin, Germany. [5]PVcomB, Helmholtz-Zentrum Berlin für Materialien und Energie GmbH, Berlin, Germany. [6]Max-Planck-Institute for Chemical Energy Conversion, Mülheim an der Ruhr, Germany. [7]Department Interface Design, Helmholtz-Zentrum Berlin für Materialien und Energie GmbH, Berlin, Germany. [8]Helmholtz Institute Erlangen-Nürnberg for Renewable Energy (HI ERN), Berlin, Germany. [9]Department of Chemistry and Pharmacy, Friedrich-Alexander-Universität Erlangen-Nürnberg (FAU), Erlangen, Germany. [10]These authors contributed equally: Zehua Li, Eylül Öztuna, Katarzyna Skorupska. ✉e-mail: skorupska@fhi-berlin.mpg.de; rs01@fhi-berlin.mpg.de

