## [Transparent Peer Review file · Nature Communications]

Rationally Designed Laterally-Condensed-Catalysts Deliver Robust Activity and Selectivity for Ethylene Production in Acetylene Hydrogenation

Corresponding Author: Dr Katarzyna Skorupska

Version 0:

Reviewer comments:

Reviewer #1

(Remarks to the Author)

In this work, the authors designed a self-repairing Pd:C system by synthesizing a laterally condensed catalyst (LCC). The functional palladium interlayer was prepared on the SiO₂ buffer layer to achieve control of the reaction interface, subsurface volume, and functional interface extending to the buffer layer. It has superior acetylene hydrogenation performance compared with conventional powder and Pd foil catalysts. Nevertheless, the manuscript still has plenty of room for improvement.

1. The LCC catalysis concept was not common in previous reports. And the new terms or concepts need to be clearly explained before they are used to justify the new terms proposed.
2. The novelty of the catalyst falls short of the journal's criteria. The LCC catalyst was not considered a groundbreaking advancement, as a similar sandwich catalyst had already been reported for hydrogenation reactions (ACS Catal. 2017, 7, 6567.).
3. In the Pd 3d XPS spectra, theoretically there should not be two completely overlapping peaks. However, the Pd0 peak fitted by the authors overlapped with the Pd:C peak completely, which requires re-fitting analysis.
4. The authors emphasize that the LCC catalyst delivers robust activity and selectivity for ethylene production in acetylene hydrogenation. However, the results of practical performance tests show that both ethylene selectivity and stability are still a certain distance from the results reported in most of the current studies.
5. The spectra of adsorbed CO reveal that only bridge-bonded CO is observed on the LCC, suggesting that the geometric structure of the catalyst is different from that of Pd foil. The authors should provide a more in-depth analysis of the geometrical coordination structure of Pd in LCC catalysts.
6. The characterization and evidence for the geometry and electronic structure of Pd in LCC catalysts in this manuscript are lacking. To provide more comprehensive insights, XANES spectroscopy is recommended to resolve the differences between LCC and Pd foil.
7. The authors compared the pretreatment effects on LCC in Ar, C₂H₂, C₂H₄ and different temperatures for selective hydrogenation of acetylene reaction. What was the main purpose of this comparison?
8. It is necessary to perform the thermogravimetric (TG) tests for the used catalysts to evaluate the possible accumulation of C₄ products.
9. The authors recommend that the activity for saturated hydrocarbon formation of the LCC film is another reason for the deactivation of catalysts. It is suggested to further confirm the formation of saturated hydrocarbons by the pyrolysis gas chromatography-mass spectrometer (GC-MS) analyses.

Reviewer #2

(Remarks to the Author)

The manuscript entitled "Laterally-Condensed-Catalysts Deliver Robust Activity and Selectivity for Ethylene Production in Acetylene Hydrogenation" have been reviewed. The authors use the multiple design criteria deduced from conventional catalysts to devise a new self-repairing Pd: C system realized via the synthesis of a laterally condensed catalyst (LCC), and

ultrahigh ethylene productivity in excess of >1 kmol C₂H₄/gPd/hour was reproducibly achieved. The catalyst material is well characterized and documented properly. Therefore, I recommend acceptance of the manuscript after major revision. Before its acceptance author should clarify the following points.

1. In the manuscript, the planar surface of Pd greatly affects the polymerization side reaction, the authors explain the reasons "Since such reactions are likely to occur on planar surface facets of Pd, it is important to design the Pd reactive interface as rough nanostructures." The authors should provide more experimental evidence to support this conclusion.
2. XRD or other characterization analysis should be used to determine the formation carbon layer between SiO₂ and Pd, instead of just relying on XPS results.
3. The authors moved the both catalysts to a quartz-bed plug flow reactor via glovebox. Are these catalysts sensitive to the oxygen and moisture in the air? If the catalyst undergoes changes, will it affect the activity results? This aspect should be very important for heterogeneous catalysts. Authors should give out the detailed explanations.
4. In practical applications, the proportion of acetylene in the acetylene-ethylene mixed gas is very low (~0.5-1.5%). It is recommended that the author should test the catalytic activity of the catalyst at this acetylene concentration.
5. There are a lot of issues about the reference format of the article, such as magazine abbreviations and so on.

Reviewer #3

(Remarks to the Author)

This manuscript reports the fabrication of Pd thin film for semihydrogenation of acetylene. These authors believe that the predeposition of carbon layer can increase the amount of subsurface carbon incorporated into Pd and therefore improve the semihydrogenation selectivity. Comprehensive characterizations were performed, together with some DFT modeling. However, the motivation and novelty of this research and the data interpretation are questionable. This manuscript might be publishable in some catalysis journals after carrying out additional experiments and significant rewriting. Detailed comments are provided below.

1. Pd/SiO₂ powder also shows comparably high ethylene productivity, but the ethylene selectivity is poor. This might be due to the presence of mesopores in the SiO₂ support they selected, which would cause deeper hydrogenation along diffusion. To this end, Pd supported on mesoporous SiO₂ may not be a good reference for comparison. In order to draw a fair comparison with their thin film catalyst that does not have diffusion limitation, they should use low surface area macroporous SiO₂ to avoid diffusion limitation.
2. All catalysts reported in this work show sharply decreased selectivity toward ethylene when the acetylene conversion approaches 100%. This is a very common behavior for many monometallic Pd nanoparticle catalysts without surface modification. The formation of Pd carbide does not seem to alter the reaction mechanism at high conversion level. The new fundamental insight provided by this work is not very clear.
3. Severe restructuring or sintering of the Pd thin film occurred upon usage. This implies that this type of catalyst is unlikely to survive through multiple regenerations under industrial relevant conditions. Is this sophisticated synthesis protocol practical?
4. The manuscript is a little difficult to read. The authors should carefully proof-read the manuscript and polish the writing before submission to a scientific journal.

Reviewer #4

(Remarks to the Author)

Version 1:

Reviewer comments:

Reviewer #1

(Remarks to the Author)

I thank the authors for their answers and additional work on the manuscript, which is surely improved. And the manuscript is of potential interest to the readers of Nature Communications.

Reviewer #2

(Remarks to the Author)

Authors have well readdressed the questions raised by the reviewer, now , the manuscript can be accepted.

Reviewer #3

(Remarks to the Author)

The authors have spent quite a bit of effort to revise the manuscript. Although I am still quite conservative about the novelty of this work, the quality of the revised manuscript has indeed improved. I don't have further comments.

Reviewer #4

(Remarks to the Author)

Reviewer #1 (Remarks to the Author):

In this work, the authors designed a self-repairing Pd:C system by synthesizing a laterally condensed catalyst (LCC). The functional palladium interlayer was prepared on the SiO₂ buffer layer to achieve control of the reaction interface, subsurface volume, and functional interface extending to the buffer layer. It has superior acetylene hydrogenation performance compared with conventional powder and Pd foil catalysts. Nevertheless, the manuscript still has plenty of room for improvement.

Answer: We thank the reviewer for the time employed in the review of our work and for the suggestions that we believe have resulted in a clearer manuscript.

1. The LCC catalysis concept was not common in previous reports. And the new terms or concepts need to be clearly explained before they are used to justify the new terms proposed.

Answer: Following the reviewer's comment, we have now expanded the section dedicated to the introduction of the LCC concept, including why this is an important new development. Traditional fundamental studies in catalysis have considered a SINGLE reactive interface. However, real working catalysts comprise four topologically different regions cooperating during active operation, namely, the top reactive interface, the underlying bulk of the same material, a functional interface and the support of a different material. Existing material science studies of catalysts largely ignore the sub-surface and functional interfaces (in contact with the support), due to the lack of sufficiently sensitive analytics or the inaccessibility of those methods to investigate buried interfaces in nanostructures. Due to its 2D layered planar design, the LCC materials platform proposed here serves to overcome such challenges.

In the LCC concept, the needed complexity of the reactive and functional interfaces present in nanoparticulate materials is retained. Nonetheless, its planar 2D-like structure helps us to make a bridge and close the materials gap existing between model single crystal catalysts (with one reactive interface and the bulk of the same material), model monolayer thin films (with the reactive interface as a monolayer and the support of a different material), and the standard 3D nanoparticulate powder industrial catalysts with the same four different regions and reactive and functional interfaces described above for the LCCs but harder to access and harder to design and stabilize (see review figure R1 below). Our new materials concept enables both, fundamental spectro-microscopy mechanistic studies and catalytic investigations. We replaced here the traditional thin films by a laterally-condensed catalyst layer made of closely packed catalyst islands, having in mind the minimization and control of subsurface regions below the reactive interface and above the functional interface with the support.

Figure Redacted

Figure R1. Evolution of the structural complexity of active catalysts ranging from model systems to performance systems. Adapted from Robert Schlögl *Arkivoc* **3** (2024) 202412178. "The functional interface in catalysis"¹.

In the original manuscript we had already attempted to explain this new concept (P3, L 9 to P5, L8). Nonetheless, we have added now also additional text to explain its rational and advantages.

Added to the manuscript, P5, L9-31:

Real working catalysts comprise four topologically different regions cooperating during active operation, namely, the top reactive interface, the underlying bulk of the same material, a functional interface and the support of a different material. Existing fundamental studies of catalysts have mainly considered a single reactive interface, largely ignoring the sub-surface and functional interface in contact with the support. This is in part due to the lack of sufficiently sensitive analytics or the inaccessibility of those methods to investigate buried interfaces in nanostructures. Due to its 2D layered planar design, the LCC materials platform proposed here contributes to overcome such challenges.

In the LCC concept, the needed complexity of the reactive and functional interfaces present in nanoparticulate materials is retained. Nonetheless, its planar 2D-like but also highly densely-packed structure closes the materials gap existing between model single crystal catalysts (with one reactive interface and the bulk of the same material), model monolayer thin films (with the reactive interface as a monolayer and the support made of a different material), and the standard 3D nanoparticulate powder industrial catalysts, the latter with the same four different regions and reactive and functional interfaces described above for the LCCs but harder to access and harder to design and stabilize. This planar 2D LCC catalyst platform serves as an intermediate stage between powder catalysts and single-crystal model systems, enabling quantitative fundamental in situ/operando spectro-microscopy studies and catalytic investigations.

We replaced here the traditional thin films by a laterally-condensed catalyst layer made of closely packed catalyst islands, having in mind the minimization and control of subsurface regions. Such materials concept is not only relevant to thermal catalysis processes but will also have exciting applications in the field of electrocatalysis, where the planar sample geometry is already being applied. The LCCs will in fact allow us to get a holistic understanding of catalytic processes by using the exactly same materials system exposed to two different activation sources (temperature vs electrical potential).

2. The novelty of the catalyst falls short of the journal's criteria. The LCC catalyst was not considered a groundbreaking advancement, as a similar sandwich catalyst had already been reported for hydrogenation reactions (ACS Catal. 2017, 7, 6567.).

Answer: We hope that with the answer to the prior comment the LCC concept is clearer. We are not dealing here with bulk-like thin film multilayers. The LCC materials platform intends to have full control of all relevant interfaces involved in a catalytic process while minimizing the amount of the typically expensive catalytic material that is used as part of the reactive interface. This is achieved here by making a reactive interface composed from nucleated islands with the highest packing density possible.

The sandwich catalyst mentioned by the reviewer in the former reference, which we have now added as reference to the revised manuscript version², is a 3D powder catalyst (see figure R2 taken from the reference) with embedded Pd nanoparticles between two porous layers of TiO₂ (TiO₂/Pt/TiO₂). The mentioned 3D configuration hinders in depth quantitative analytic studies with surface science spectro-microscopy tools as the ones employed here. Moreover, in the former study the dispersion of the embedded particles could not be controlled, as it is the case here, where the packing (inter-island distance) of the Pd nuclei serving as precursor for a thin film can be tuned.

In the mentioned study, the ALD synthesis of TiO₂/Pt/TiO₂ catalysts with a tubular sandwich structure by ALD was described. Pt nanoparticles were there confined between two porous TiO₂ layers. The superior catalytic performance of the mentioned system for tandem ammonia–borane decomposition was ascribed to the lack of Pt metal surface sites, with only Pt–TiO₂ interface sites available.

The Pd LCC design presented here includes a reactive interface and a non-reactive functional interface. This 2D LCC catalyst serves as an intermediate stage between powder catalysts and single-crystal model systems, enabling both fundamental studies and catalytic investigations.

Figure Redacted

Figure R2, Abstract figure from the reference²

On the other hand, the Pd LCC design presented here is planar (2D-like), better accessible for characterization and it includes a reactive interface and a non-reactive functional interface that can be both separately tuned. This 2D LCC catalyst serves as an intermediate stage between powder catalysts and single-crystal model systems, enabling both fundamental studies and catalytic investigations.

In addition to the new text added as response to the first comment, we have also added the following new text:

Added to the manuscript, P5, L22-25:

This planar 2D LCC catalyst platform serves as an intermediate stage between powder catalysts and single-crystal model systems, enabling quantitative fundamental in situ/operando spectro-microscopy studies and catalytic investigations.

3. In the Pd 3d XPS spectra, theoretically there should not be two completely overlapping peaks. However, the Pd⁰ peak fitted by the authors overlapped with the Pd:C peak completely, which requires re-fitting analysis.

Answer: There must be some misunderstanding here. Our team has very extensive expertise in the analysis of XPS spectra since we operate ourselves several beamlines including this technique at our local synchrotron BESSY II.

It is well known, that the chemical shift induced by C in metals like Pd or Ni is small. Therefore, the degree of overlapping depends on the broadening of the core level. There are a lot of examples in the literature showing the overlap that the referee addressed Pd-C^{3,4}. The overlap would be even bigger when a lab source XPS would have been used to measure the spectrum. The overlap is also more evident when broad spectral features are detected, as it is typical for cases such as the present one, when the reactive interface under investigation is made of small but closely-packed nucleation islands. Thus, we are confident that our fits (Figure 4b, Figure 5a and Figure S17a) are adequate.

4. The authors emphasize that the LCC catalyst delivers robust activity and selectivity for ethylene production in acetylene hydrogenation. However, the results of practical

performance tests show that both ethylene selectivity and stability are still a certain distance from the results reported in most of the current studies.

Answer: It seems that the reviewer has missed the detailed comparison that we have included in the original manuscript in Table S9. There we present an overview of catalytic results from the literature and compare them with the results obtained for our Pd LCC. It should be emphasized that this comparison is challenging due to variations in catalytic parameters and catalyst quantities across different studies, where we have selected the closest catalyst loadings and conditions that we could find in the literature. In fact, our Pd LCC shows comparable selectivity but significantly higher ethylene productivity as compared to the powder catalysts reported in the literature. Below we copy a modified version of the sentence that the reviewer mentioned for clarity.

Added to the manuscript, P11, L20-22:

With a comparable selectivity, our productivity is outstanding and at least an order of magnitude better than many benchmark powder systems, as indicated in Table S9 for similar reaction conditions.

5. The spectra of adsorbed CO reveal that only bridge-bonded CO is observed on the LCC, suggesting that the geometric structure of the catalyst is different from that of Pd foil. The authors should provide a more in-depth analysis of the geometrical coordination structure of Pd in LCC catalysts.

Answer: CO is a probe molecule that can probe the electronic and geometrical structure of metal surfaces extremely sensitively. The fact that only CO adsorbed in a bridged coordination over two palladium atoms is observed on the LCC catalyst indicates that the surface is very rough. We conclude this on the basis of a comparison of our spectra with published data⁵ in the submitted manuscript. A rough, stepped Pd surface was observed in our cross-section electron microscopy data (Fig. R3, now new Suppl. Fig. S20b) (see also Figure 4g in the originally submitted manuscript). The observed undefined polyhedral shapes, along with the absence of larger facets, can explain the absence of highly coordinated surface sites. The expected roughness of the Pd foil occurs on a different length scale (microns) as compared to the nanometer scale of the LCC.

Figure R3, Cross-section HRTEM micrograph of Si – 20 nm SiO₂ – 3 nm Pd.

To make this clearer, we have improved the explanation of the TEM image in the revised manuscript and Figure R3 has been added to the Fig. S20b in the revised manuscript:

Added to the manuscript, P9, L10-13:

By tilting the FIB lamella by 15 deg, the granular isotropic structure of the LCC becomes visible (Figure 4g, top, Figure S20b), resulting in a rough, stepped surface. This is in accord with our IR data unveiling the presence of low-coordinated surface sites that favor the binding of CO in bridge-mode configuration (Figure S29).

In addition, we also refer to the electron micrographs in the revised manuscript when discussing the IR spectra:

Added to the manuscript, P10, L16-19:

Using CO as probe molecule for the Pd state after the standard hydrogen activation, the activated surface appears rough and stepped, which is in agreement with TEM (Figure 4g, top, and Figure S20b), and far from the observations previously made for GaPd systems⁶ that are characterized by a large faceted state.

Added Figure to SI: Figure S20b

6. The characterization and evidence for the geometry and electronic structure of Pd in LCC catalysts in this manuscript are lacking. To provide more comprehensive insights, XANES spectroscopy is recommended to resolve the differences between LCC and Pd foil.

Answer: We have included in the original manuscript data from the NEXAFS Pd L-edge for Pd LCC (Fig 5b). Following the recommendation from the reviewer, we have now added additional data from a Pd foil as comparison and the corresponding explanations (Fig. R4, now new Suppl. Fig. S14b). The figure shows a comparison of Pd L₃-edge NEXAFS spectra measured in TEY (total electron yield) mode. The differences between the two spectra are the slight shift

in the whiteline energy towards more positive values (0.2 eV) and the broadening of the whiteline peak in Pd LCC as compared to that of the Pd foil. Both changes might result from a particle size effect, which was previously reported when comparing Pd L₃-edge spectra of a Pd foil with those from particles smaller than 10 nm⁷. Another explanation for the difference in the NEXAFS Pd L₃-edge spectra might be the different amounts of carbon incorporated into the Pd lattice. As it is demonstrated in the manuscript, with increasing the amount of carbon in the Pd lattice, the whiteline shifts to higher energies and broadens (Figure 5b). The absence of surface or bulk Pd oxide in the Pd LCC is already revealed by the XPS Pd 3p/O 1s core level fittings and comparison of the spectra with that of reference measurements performed with a Pd foil (Figure S15a). The absence of Pd oxide is also validated by the Pd L₃-edge spectrum of Pd LCC, since the whiteline of PdO has a significantly higher energy (by 1.3 eV) and the line shape is also more symmetric than that of metallic Pd, which is not the case for our Pd LCC sample⁸.

Figure R4, Pd L₃-edge spectra of Si – 20 nm SiO₂ – 3 nm Pd (Pd LCC) and Pd foil measured at room temperature and UHV. NEXAFS was measured at the synchrotron BESSY II.

Added to SI, P32, L39 to P33, L7:

Figure S14b shows a comparison of Pd L₃-edge NEXAFS spectra of a fresh Pd LCC (Si – 20 nm SiO₂ – 3 nm Pd) sample and Pd foil. The differences between the two spectra are the slight shift in the whiteline energy towards more positive values (0.2 eV) and the broadening of the whiteline peak in the Pd LCC sample as compared to that of the Pd foil. Both changes might result from a particle size effect, which was previously reported when comparing Pd L₃-edge spectra of a Pd foil with those from particles smaller than 10 nm⁷. Another explanation for the difference in the NEXAFS Pd L₃-edge spectra might be the different amounts of carbon incorporated into the Pd lattice. As it is demonstrated in the manuscript, with increasing the amount of carbon in the Pd lattice, the whiteline shifts to higher energies and broadens (Figure 5b). The absence of surface or bulk Pd oxide in the Pd LCC is already revealed by the XPS Pd 3p/O 1s core level fittings and comparison of the spectra with that of reference measurements performed with a Pd foil (Figure S15a). The absence of Pd oxide is also validated by the Pd L₃-edge spectrum of Pd LCC, since the whiteline of PdO has a significantly higher energy (by 1.3 eV) and the line shape is also more symmetric than that of metallic Pd, which is not the case for our Pd LCC sample⁸.

Added to the manuscript, P9, L44-47:

The fresh sample has spectral features assigned to metallic Pd with a slightly shifted and broader whiteline compared to that of the Pd foil, due to differences in the particle size and carbon content (Figure S14b)⁷. During the reaction at 100 °C, the whiteline shifts to higher energies and broadens which is a distinct signature of Pd:C phase under alkyne hydrogenation conditions⁹.

Added Figure to SI: Figure S14b

7. The authors compared the pretreatment effects on LCC in Ar, C₂H₂, C₂H₄ and different temperatures for selective hydrogenation of acetylene reaction. What was the main purpose of this comparison?

Answer: It was known from the literature that the interaction of Pd with carbon plays a role in this reaction. Our pre-treatment study aimed to explore whether the source or the way we introduce carbon into the system influences its activity and selectivity. In particular, the different pre-treatments were expected to lead to a different content of carbon incorporated into Pd or on the Pd surface. Such different starting pre-catalysts were then compared to draw conclusions on the optimum pre-catalyst composition and state.

Here, in order to establish the optimal pretreatment protocol, several temperatures (Fig. S27) and gas feeds (Fig. S26) were tested. Additionally, the catalytic performance of the Pd LCC without any pretreatment was evaluated (Fig. S27). The purpose of studying the catalytic performance of samples pretreated in gas feeds with and without a carbon source was to understand the influence of carbon incorporation into the Pd lattice and carbon blocking of the Pd surface. Surface blocking of the Pd LCC is particularly evident in the case of acetylene pretreatment, which leads to low conversion values (Fig. S26a). Our pre-treatment study revealed that control over the amount of atomic carbon entering the Pd subsurface can be achieved by introducing a known amount of carbon during LCC synthesis (Fig. 3b-f) and by using a carbon-based feed gas during sample pretreatment/activation (Fig. S26). We have now better clarified this aspect in the manuscript text.

Original text in the manuscript: P9, L39-40:

The improvement arises from adapting the activation gas composition such as to maximise the Pd:C content (Figure S26).

Added to SI, P51, L21-29:

By comparing different catalyst pre-treatments we were able to gain insight into whether the source or the way we introduce carbon into the Pd LCC influences its activity or selectivity. In particular, different pre-treatments are expected to result in a different content of carbon incorporated into Pd or deposited on the Pd surface, which would either contribute to the stabilization of the material or to the blocking of active sites. Such different starting pre-catalysts conditions were then compared to draw conclusions on the optimum pre-catalyst composition and state. Here we observed that an improvement in the catalytic performance arises when the gas composition during the activation process is adjusted such as to maximize the Pd:C content (Figure S26).

8. It is necessary to perform the thermogravimetric (TG) tests for the used catalysts to evaluate the possible accumulation of C₄ products.

Answer: We agree with the reviewer that thermal analysis is generally a suitable method for determining the quantity and type of carbon deposits on spent catalysts. However, we would like to point out that such an analysis is significantly more challenging on the thin films than on a normal nanocrystalline powder catalyst due to the low specific surface area of the LCC samples. Nevertheless, at the suggestion of the reviewer, we conducted thermogravimetric analyses coupled with mass spectrometry (TG-MS) and included a new figure R5e,f (now new Suppl. Figure S23e,f) in the revised manuscript. The mass loss and the TG-MS signal for CO₂ ($m/z = 44$) were compared for the as-prepared Pd LCC sample and the spent catalyst after 10h time on stream (TOS). The measurements were performed in synthetic air (21% O₂ in Ar). We attribute the initial mass increase recorded on both, the as-prepared and spent LCC catalysts in the temperature range 40-150°C to oxidation of the thin Pd film. While the as-prepared LCC shows no mass loss at higher temperatures, we attribute the mass loss of 0.01% up to 700°C to the decomposition and combustion of carbonaceous deposits on the spent LCC, in agreement with the MS signal for $m/z = 44$. The different maxima of the MS signal at 350°C, 500°C and 650°C indicate that different types of carbonaceous species are present on the surface. In agreement with the Raman spectra of the spent sample (see our response to comment no. 9 from the reviewer), these species can be attributed to molecular carbonaceous species, deposited polyacetylene and perhaps to a minor extent to green oil¹⁰. The formation of graphitic carbon does not occur. Temperature Programmed Desorption (TPD) performed in an Ar atmosphere on the spent Pd LCC sample (TOS = 10h) showed no signal $m/z = 27$ (Figure R5g), which would be expected as a significant fragment in the mass spectra of adsorbed shorter hydrocarbons, like C₂-C₄. In contrast, the contribution of fragment 27 is much less in the mass spectrum of green oil, which is composed of linear C₁₂ – C₃₀ hydrocarbons. Therefore, the absence of the m/z signal 27 suggests that the carbon deposits on the catalyst surface consist of heavy hydrocarbons or polymerized chains that can only be removed in an oxidizing atmosphere (Fig. R5e,f). We have introduced the following modifications in the revised manuscript:

Added to the manuscript, P9, L33-38:

The formation of carbonaceous deposits, as evidenced by the carbon balance, thermal analysis, and Raman spectroscopy (Figure S23), is not the only reason for deactivation, the activity for saturated hydrocarbon formation of the LCC film is another reason, although being lower than that of dispersed Pd/SiO₂ (Figure S24g). The accumulation of carbon deposits on the sample surface after extended operation time (260 h) was revealed by Raman spectroscopy (Figure S23).

Added to SI, P49, L11-30:

As shown in Figure S23e,f, the mass loss and the TG-MS signal for CO₂ ($m/z = 44$) were compared for the as-prepared and the spent Pd LCC samples after 10h time on stream (TOS). The measurements were performed in synthetic air (21% O₂ in Ar). We attribute the initial mass increase recorded on both, the as-prepared and spent LCC catalysts in the temperature range 40-150°C to oxidation of the thin Pd film. While the as-prepared LCC shows no mass loss at higher temperatures, the mass loss of 0.01% up to 700°C is likely assigned to the decomposition and combustion of carbonaceous deposits on the spent LCC, in agreement with the MS signal for $m/z = 44$. The different maxima of the MS signal at 350°C, 500°C and 650°C indicate that different types of carbonaceous species are present on the surface. In agreement with the Raman spectra of the spent sample (Figure S23c,d), these species can be attributed

to molecular carbonaceous species, deposited polyacetylene and perhaps to a minor extent to green oil¹⁰. The presence of graphitic carbon species was not detected. Temperature Programmed Desorption (TPD) performed in an Ar atmosphere on the spent Pd LCC sample (TOS = 10h) showed no signal for $m/z = 27$ (Figure S23g), which would otherwise be expected as a significant fragment in the mass spectra of adsorbed shorter hydrocarbons, like C2-C4. In contrast, the contribution of fragment 27 is much lower in the mass spectrum of green oil, which is composed of linear C12 – C30 hydrocarbons. Therefore, the absence of the m/z signal 27 suggests that the carbon deposits on the catalyst surface consist of heavy hydrocarbons or polymerized chains that can only be removed in an oxidizing atmosphere (Figure S23e,f).

Added Figure to SI: Figure S23.

Figure R5, Observation of carbon deposits. (a) carbon balance of the catalytic acetylene hydrogenation on LCC Pd, Pd powder and Pd foil. (b) mass spectra (analog scan) on the product gas on LCC Pd at TOS = 66h. (c) In-situ Raman spectra of "green oil" produced in the Raman cell (black, TOS=27h) and carbon deposits on spent LCC Pd after TOS=260h (red). (d) Carbon deposits observed by Raman spectroscopy for fresh, spent Pd LCCs (TOS= 49h, 260h). (e) TG and (f) MS observation on fresh and spent LCC Pd in synthetic air (21% O₂ in Ar) and (g) inert Ar. Temperature was increased from RT to 700°C at rate of 10°C/min in TG-MS. (h-j) Light microscopy images obtained by Raman for the fresh, spent Pd LCCs (TOS= 49h, 260h). Reaction condition: (a) C₂H₂: 0.9 ml/min, H₂: 27 ml/min, N₂: 5.8 ml/min, T = 150°C and the spent LCC in

(c-g, i, j) for Raman; (b) C₂H₂: 0.9 ml/min, H₂: 0.9 ml/min, N₂: 31.9 ml/min, T = 150°C; and (c) C₂H₂: 1.5 ml/min, H₂: 1.5 ml/min, N₂: 27 ml/min, T = 150°C for in-situ Raman.

9. The authors recommend that the activity for saturated hydrocarbon formation of the LCC film is another reason for the deactivation of catalysts. It is suggested to further confirm the formation of saturated hydrocarbons by the pyrolysis gas chromatography-mass spectrometer (GC-MS) analyses.

Answer: C₄ products were detected in the product gas mixture of the catalytic tests using online gas chromatography as reported in the original manuscript (Figures 3e, 4i, S21a, S24a-d, S30a iv, S30b iv). Additionally, we performed product detection using online mass spectrometry (Figure R5b, new Suppl. Figure S23b). The following m/z intensities were detected: H₂ (m/z=2), N₂ (m/z=14 and 28), C₂H₂ (m/z=26), C₂H₄ (m/z=28), C₂H₆ (m/z=30), C₄H₆ (m/z=39, 54) C₄H₈ (m/z=41, 56), C₄H₁₀ (m/z=43, 58). Other minor fragments/peaks were also found. As expected, heavier hydrocarbons were not detected in the gas phase. We have now added the new Figure S23 in the revised Supporting Information to clarify this aspect.

Higher hydrocarbons remain adsorbed on the catalyst surface, on the reactor walls and in the tubing. We have now used TG-MS in Ar (see reply to Point 8 and the Figure R5g) and Raman spectroscopy to analyze the nature of the deposits on the spent catalyst. A new Figure R5d showing the Raman spectra of several spent LCC samples after different times on stream (TOS 49 hours and 260 hours) was included in the Supporting Information. The Raman spectrum of the as-prepared LCC is essentially free of features due to carbon, although a weak broad band close to 1600 cm⁻¹ suggests the presence of minor graphitic impurities in the fresh catalyst. Two strong peaks at ca. 1100 cm⁻¹ and ca. 1500 cm⁻¹ can be detected in the spent catalysts, which are attributed to the characteristic modes of carbon-carbon single bond and carbon-carbon double bond stretching vibrations, respectively, in adsorbed *trans*-polyacetylene¹¹. The frequency of the C=C stretching mode depends on the structure and the conjugation length of the polymer¹². C-H stretching vibrations are also observed at 2920 cm⁻¹. The coverage of the surface with polyacetylene is also in agreement with the thermal analysis in air (see our reply to comment 8 and Figure R5e,f) as polyacetylene undergoes a phase transition and oxidative degradation at temperatures above 300°C¹³. Images taken from the camera of the Raman microscope show also different light refraction for the sample after 49 and 260 hours TOS, respectively, suggesting the presence of different thicknesses of organic material on the surface of these two samples. As the composition of the organic film and therefore the refractive index is not precisely known, the color cannot be used to determine the exact film thickness.

The Raman spectrum of an oily liquid ("green oil") recovered from the cell walls of the Raman cell in an operando experiment in acetylene hydrogenation is shown in the new Figure R5c. The spectrum differs from the spectrum of the carbonaceous deposits on the used catalysts, so that it can be assumed that the organic layer on the used catalysts consist mainly of polyacetylene and not of the green oil.

Revised text (see answer to point 8) has been added to the manuscript. Figure R5 has been added to SI (Figure S23).

Further detailed discussion has been added to SI, P49, L31 to P50, L9:

We performed product detection using online mass spectrometry (Figure S23b). The following m/z intensities were measured: H_2 ($m/z=2$), N_2 ($m/z=14$ and 28), C_2H_2 ($m/z=26$), C_2H_4 ($m/z=28$), C_2H_6 ($m/z=30$), C_4H_6 ($m/z=39, 54$) C_4H_8 ($m/z=41, 56$), C_4H_{10} ($m/z=43, 58$). Other minor fragments/peaks were also found. As expected, heavier hydrocarbons were not detected in the gas phase. Higher hydrocarbons remain adsorbed on the catalyst surface, on the reactor walls and in the tubing. Raman spectroscopy was used to analyze the nature of the deposits on the spent catalyst. The Raman spectrum of the as-prepared LCC is essentially free of features due to carbon, although a weak broad band close to 1600 cm^{-1} suggests the presence of minor graphitic impurities in the fresh catalyst (Figure S23d). Two strong peaks at ca. 1100 cm^{-1} and ca. 1500 cm^{-1} can be detected in the spent catalysts, which are attributed to the characteristic modes of carbon-carbon single bond and carbon-carbon double bond stretching vibrations, respectively, in adsorbed trans-polyacetylene¹¹. The frequency of the C=C stretching mode depends on the structure and the conjugation length of the polymer¹². C-H stretching vibrations are also observed at 2920 cm^{-1} . The coverage of the surface with polyacetylene is also in agreement with the thermal analysis in air (Figure S23e,f) as polyacetylene undergoes a phase transition and oxidative degradation at temperatures above 300°C ¹³. Images taken from the camera of the Raman microscope show also different light refraction for the sample after 49 and 260 hours TOS, suggesting the presence of different thicknesses of organic material on the surface of these two samples. As the composition of the organic films and therefore the refractive index is not precisely known, the color cannot be used to make an estimation of the film thickness. The Raman spectrum of an oily liquid ("green oil") recovered from the cell walls of the Raman cell in an operando acetylene hydrogenation experiment is shown in Figure S23c. The spectrum differs from the spectrum of the carbonaceous deposits on the used catalysts, so that it can be assumed that the organic layer on the used catalysts consist mainly of polyacetylene and not of the green oil.

Reviewer #2 (Remarks to the Author):

The manuscript entitled "Laterally-Condensed-Catalysts Deliver Robust Activity and Selectivity for Ethylene Production in Acetylene Hydrogenation" have been reviewed. The authors use the multiple design criteria deduced from conventional catalysts to devise a new self-repairing Pd: C system realized via the synthesis of a laterally condensed catalyst (LCC), and ultrahigh ethylene productivity in excess of $>1\text{ kmol } C_2H_4/g_{Pd}/\text{hour}$ was reproducibly achieved. The catalyst material is well characterized and documented properly. Therefore, I recommend acceptance of the manuscript after major revision. Before its acceptance author should clarify the following points.

Answer: We Thank the reviewer for the time dedicated to the evaluation of our manuscript, and the positive constructive feedback provided, which has resulted in an improvement if its clarity.

1. In the manuscript, the planar surface of Pd greatly affects the polymerization side reaction, the authors explain the reasons "Since such reactions are likely to occur on planar surface facets of Pd, it is important to design the Pd reactive interface as rough nanostructures." The authors should provide more experimental evidence to support this conclusion.

Answer: We based our claim on knowledge previously reported in the literature¹⁴⁻¹⁶ as well as experience we have gathered from confidential industrial collaboration work.

However, experimental studies on the dependence of geometry vs. polymerization side reaction were not in the scope of this work. Nonetheless, our TG-MS results showing extremely low mass loss (Figure R5e) point in the direction of a reduced amount of carbon deposits on the rough surface of Pd LCC.

It was reported in literature that a suppression of the side reactions can be obtained by design of the surface morphology with metal atoms having low coordination numbers (edge and corner atoms). For instance, in theoretical studies of flat Pd(111) and stepped Pd(211) surfaces it was found that the favorable site for C₄H₆ formation is at the former and not at the latter surface. DFT calculations show that an ensemble of 4 sites of Pd is required to permit acetylene oligomerization.

Added to the manuscript, P6, L7-10:

Since such reactions are likely to occur on planar surface facets of Pd, it is important to design the Pd reactive interface as rough nanostructures^{14,15} with small Pd ensembles containing a large number of edge and corner atoms being crucial to avoid oligomerization side reactions of acetylene.

2. XRD or other characterization analysis should be used to determine the formation carbon layer between SiO₂ and Pd, instead of just relying on XPS results.

Answer: The content of carbon in Si(100)- 20 nmSiO₂-0.4 nm C- 3 nm Pd was also studied with TEM lamella cross section. Figure R6 (now new Suppl. Fig. S7f,g) shows an overlap of the Pd LCC signal with carbon.

In Figure R6a we resolve the Pd:C LCC of approximately 2.5 nm by cross sectional analysis via STEM (Scanning Transmission Electron Microscopy). The sample was prepared by FIB (Focused Ion Beam) and analyzed in STEM oriented to 101 zone axis direction of the Si substrate to resolve the interfaces shapely. Figure R6b shows EDX (Energy-Dispersive X-ray spectroscopy) elemental maps of Si (light blue), O (red), C (dark blue) and Pd (yellow) and their according elemental profiles extracted from the EDX maps from top to bottom, integrated in a horizontal direction. The dark blue and yellow peak in the profiles (see arrow in Figure R6b, right) point out the C and Pd elemental distribution across the Pd:C LCC, respectively.

Added to the manuscript, P7, L9-10 (We already had some text in the revised manuscript regarding our controlled placement of the carbon layer in our reference sample that we have now further clarified):

The presence of carbon in the as prepared Pd LCC sample was also confirmed with TEM lamella cross section studies (Figure S7g, f).

Added to SI, P20, L6-13:

In Figure S7f we studied the Pd:C LCC of approximately 2.5 nm by cross sectional analysis via STEM (Scanning Transmission Electron Microscopy). The sample was prepared by focused ion beam (FIB) and analyzed in STEM oriented to 101 zone axis direction of the Si substrate to resolve the interfaces sharply. Figure S7g shows Energy-Dispersive X-ray spectroscopy (EDX) elemental maps of Si (light blue), O (red), C (dark blue) and Pd (yellow) and their corresponding elemental profiles extracted from the EDX maps from top to bottom, integrated in horizontal

direction. The dark blue and yellow peaks in the profiles (see arrow in Figure S7g, right) point out the C and Pd elemental distribution across the Pd:C LCC.

Added Figure to SI: Figure S7.

Figure R6, Cross section analysis of an as prepared Pd:C thin film by STEM. (a) STEM-HAADF image of a Pd:C thin film on top of SiO₂ and Si (left image). The Si substrate was oriented to 101 zone axis direction. An additional electron beam-deposited SiO_x layer on top of the Pd:C surface was used as protection layer for the cross section preparation by FIB. Magnified view of the Pd:C thin film (right, top) and Si substrate (bottom, right). (b) STEM-EDX maps of Si, O, C and Pd of the Pd:C thin film on top of the SiO_x and Si, and the according elemental profiles extracted from the EDX maps.

3. The authors moved the both catalysts to a quartz-bed plug flow reactor via glovebox. Are these catalysts sensitive to the oxygen and moisture in the air? If the catalyst undergoes changes, will it affect the activity results? This aspect should be very important for heterogeneous catalysts. Authors should give out the detailed explanations.

Answer: Yes, the Pd samples are susceptible to oxidation and hydroxylation upon air and moisture exposure, but the pre-treatment carried out in the reactor can serve to at least partially offset this change in the surface composition, although the morphology of the pre-catalyst would also be affected. Nonetheless, the main problem that we wanted to avoid here using the glovebox was to protect the UHV-prepared references samples from adventitious carbon exposure, since we wanted to have a well-defined Pd content in our samples,

especially those prepared by PVD with a carbon underlayer. No protective atmosphere was required for the other samples.

The text is expanded and moved from the main text to Method section in the manuscript, P14, L1-4:

The UHV-prepared Pd/C/SiO₂ samples were moved via glovebox transfer to a quartz-bed plug flow reactor and tested for acetylene hydrogenation at 150°C. This inert sample transfer was performed in order to prevent the modification of the well-defined carbon content purposely deposited in these samples.

and P16, L19-22

Only the UHV-deposited model Pd LCC samples with a defined amount of carbon were transferred to the reactor using an inert transfer method. The remaining samples were transferred in air, after confirming that this transfer did not affect their characteristics.

4. In practical applications, the proportion of acetylene in the acetylene-ethylene mixed gas is very low (~0.5-1.5%). It is recommended that the author should test the catalytic activity of the catalyst at this acetylene concentration.

Answer: There seems to be a misunderstanding here that we are now addressing. The scope of this work was not to study the industrial process where cleaning ethylene stream from a few percent acetylene in required. Chemical reduction of CO₂ while generating valuable basic chemicals will co-generate ethylene and acetylene by plasma pyrolysis (**Figure 1**). The acetylene fraction (much higher than industrial cleaning process, see reference¹⁷) should be selectively hydrogenated to valuable ethylene to eliminate operational dangers of the desired low-temperature carbon formation.

We had already some text on this in the original manuscript that we have now modified:

Revised in the manuscript, P2, L8-13:

Such reactions leading to black carbon will co-generate ethylene and acetylene by plasma pyrolysis. The highly concentrated acetylene fraction¹⁷ should be selectively hydrogenated to valuable ethylene to eliminate operational dangers of the desired low-temperature carbon formation. In concentrated gas streams, the exothermic semi-hydrogenation¹⁸ is more demanding than the thermodynamically preferred full hydrogenation and thus, the formation of polymeric carbon must be suppressed. (Suppl. Note 1).

5. There are a lot of issues about the reference format of the article, such as magazine abbreviations and so on.

Answer: The format of the references has been corrected.

Reviewer #3 (Remarks to the Author):

This manuscript reports the fabrication of Pd thin film for semihydrogenation of acetylene. These authors believe that the predeposition of carbon layer can increase the amount of subsurface carbon incorporated into Pd and therefore improve the semihydrogenation selectivity. Comprehensive characterizations were performed, together with some DFT

modeling. However, the motivation and novelty of this research and the data interpretation are questionable. This manuscript might be publishable in some catalysis journals after carrying out additional experiments and significant rewriting. Detailed comments are provided below.

Answer: We thank the reviewer for the time dedicated to evaluating our study but we respectfully disagree with the claim of lack of sufficient novelty to be published in Nature Comm. From the reviewer's comments it was evident to us that we were not sufficiently clear on what are the main novel aspects of this work, and thus, we have now added the following next text to the paper to address this key criticism:

Added to the manuscript, P5, L9-31:

Real working catalysts comprise four topologically different regions cooperating during active operation, namely, the top reactive interface, the underlying bulk of the same material, a functional interface and the support of a different material. Existing fundamental studies of catalysts have mainly considered a single reactive interface, largely ignoring the sub-surface and functional interface in contact with the support. This is in part due to the lack of sufficiently sensitive analytics or the inaccessibility of those methods to investigate buried interfaces in nanostructures. Due to its 2D layered planar design, the LCC materials platform proposed here contributes to overcome such challenges.

In the LCC concept, the needed complexity of the reactive and functional interfaces present in nanoparticulate materials is retained. Nonetheless, its planar 2D-like but also highly densely-packed structure closes the materials gap existing between model single crystal catalysts (with one reactive interface and the bulk of the same material), model monolayer thin films (with the reactive interface as a monolayer and the support made of a different material), and the standard 3D nanoparticulate powder industrial catalysts, the latter with the same four different regions and reactive and functional interfaces described above for the LCCs but harder to access and harder to design and stabilize. This planar 2D LCC catalyst platform serves as an intermediate stage between powder catalysts and single-crystal model systems, enabling quantitative fundamental in situ/operando spectro-microscopy studies and catalytic investigations.

We replaced here the traditional thin films by a laterally-condensed catalyst layer made of closely packed catalyst islands, having in mind the minimization and control of subsurface regions. Such materials concept is not only relevant to thermal catalysis processes but will also have exciting applications in the field of electrocatalysis, where the planar sample geometry is already being applied. The LCCs will in fact allow us to get a holistic understanding of catalytic processes by using the exactly same materials system exposed to two different activation sources (temperature vs electrical potential).

1. Pd/SiO₂ powder also shows comparably high ethylene productivity, but the ethylene selectivity is poor. This might be due to the presence of mesopores in the SiO₂ support they selected, which would cause deeper hydrogenation along diffusion. To this end, Pd supported on mesoporous SiO₂ may not be a good reference for comparison. In order to draw a fair comparison with their thin film catalyst that does not have diffusion limitation, they should use low surface area macroporous SiO₂ to avoid diffusion limitation.

Answer: We would like to emphasize that the comparison with the powder systems is not a main component of the present study, but simple intended as benchmarking. It should also be noted that it is impossible to find in the literature prior studies carried out under exactly the same reaction conditions and for comparable metal loadings as the ones employed in our LCC approach. This being said, we synthesized some reference samples with typical methods proven in the prior literature. Describing the role of the porosity of the oxide support, although an important point for other studies on powder catalysts, is thus outside the main scope of the present fundamental study focused on the planar LCC samples. We take however this interesting suggestion from the reviewer for future benchmarking work.

2. All catalysts reported in this work show sharply decreased selectivity toward ethylene when the acetylene conversion approaches 100%. This is a very common behavior for many monometallic Pd nanoparticle catalysts without surface modification. The formation of Pd carbide does not seem to alter the reaction mechanism at high conversion level. The new fundamental insight provided by this work is not very clear.

Answer: We would like to clarify that we do not observe any Pd carbide formation. In fact, we report here Pd LCC structures with dissolved atomic carbon into the Pd lattice (Fig. 4b, 5a,b), in agreement with prior literature report on the reaction mechanism of subsurface carbon in Pd by our group before³. The amount of atomic subsurface carbon increases during the operando measurements, indicating its role in the reaction mechanism and materials' stability (Fig. 4b, 5a,b).

In this study, we design successfully this kind of subsurface carbon in a well-controlled manner by both, UHV (4.4×10^{-9} mbar) preparation of a very thin carbon sublayer underneath a carbon-free Pd film, as well as a sputtered (4×10^{-3} mbar) Pd LCC sample having adventitious carbon incorporated during the deposition. The subsurface carbon (Pd:C) in both samples is self-repairing during acetylene hydrogenation to have superior catalytic performance (conversion, selectivity and productivity) which is comparable to powder catalyst but much less catalyst materials is in Pd LCC. This is our new finding and it can be realized by synthesizing a laterally condensed. Such design can further bridge the fundamental study on model catalyst and practical catalytic study on benchmark systems. This first highlight paper presents the concept of the LCC catalyst now better described in the revised manuscript.

We highlight below some text that already was part of the original manuscript and further supplement some new fundamental insights on the behavior of carbon in LCC:

Revised In the manuscript, P5, L43 to P6, L2:

Substantial filling of the Pd lattice with carbon is thermodynamically favorable and the Pd:C system is self-repairing. Figure S1 predicts that the energy gain for carbon dissolution vanishes only just above 1/3 filling of the octahedral voids at the Pd(111) subsurface. Theory thus supports the concept that a spontaneous reaction between Pd and C can create, in a self-limited fashion, the optimal compromise between activity and selectivity. This motivates us to experimentally prepare first a model catalyst with a well-controlled subsurface carbon in a clean UHV system and later a benchmark system which can be realized in a practical environment.

and P7, L29-36:

The model experiments confirm the validity of the design assumptions and highlight that both, the carbon concentration and the nature of the functional interface need attention in combination with the hydrogen chemical potential of the reaction. In this study, we focus on the rational design of this kind of subsurface carbon in a well-controlled manner by both, an UHV preparation of a very thin carbon sublayer, as well as by sputtering Pd LCC in a carbon-rich (adventitious) environment. The subsurface carbon (Pd:C) in both samples is self-repairing during acetylene hydrogenation, which leads to an excellent catalytic performance (conversion, selectivity and productivity) comparable to that of powder catalyst but using much less catalyst material.

and P9, L24-28:

At the start of the catalytic test, with conversion near 100%, the selectivity toward undesired ethane (C₂H₆) exceeds that of ethylene (C₂H₄). Over time, this trend reverses, with a more pronounced improvement for Pd LCC as compared to bulk materials (Figure S30). This catalyst formation phase involves atomic carbon entering octahedral sites in the Pd lattice promoting selective acetylene hydrogenation as predicted by DFT^{19,20}.

3. Severe restructuring or sintering of the Pd thin film occurred upon usage. This implies that this type of catalyst is unlikely to survive through multiple regenerations under industrial relevant conditions. Is this sophisticated synthesis protocol practical?

Answer: In this work, we present the concept of a 2D-planar LCC catalyst and a first attempt of its realization. The reviewer is correct that the system as it is present not yet optimized, but we believe that enhanced stability can be achieved via the rational design of an improved functional interface that could serve to minimize restructuring or sintering. This is however beyond the scope of this first landmark publication. It should be however emphasized that it was our aim to be able to develop a catalyst design and synthesis platform that would be scalable and amenable for industrial implementation. This is in fact one of the highlights of our approach, that we were able to use thin film synthesis technology already demonstrated and industrially used since numerous years for the growth of solar cell materials. The synthesis protocol, based on a well-established methodology commonly used in photovoltaics, is not considered highly complex. Additionally, scalability is one of the key advantages of this fabrication procedure.

Additionally, optimizing the regeneration protocol could further mitigate these concerns. It is important to note that the formation and steady-state phases of the catalyst are highly dynamic, and morphological changes during catalytic operation are expected. This is the subject of ongoing work from our team.

4. The manuscript is a little difficult to read. The authors should carefully proof-read the manuscript and polish the writing before submission to a scientific journal.

Answer: Following the Reviewer's comment, we have now added new explanatory text and carefully revised the existent draft to make sure that it is clearer for a broader audience.

Reviewer #4 (Remarks to the Author):

Answer: We thank the reviewer for co-reviewing our study and for the suggestions that we believe have resulted in a clearer manuscript.

References

- 1 Schlögl, R. The functional interface in catalysis. *Arkivoc* **2024**, 202412178 (2024). doi:10.24820/ark.5550190.p012.178
- 2 Liang, H. et al. Porous TiO₂/Pt/TiO₂ sandwich catalyst for highly selective semihydrogenation of alkyne to olefin. *ACS Catal.* **7**, 6567–6572 (2017). doi:10.1021/acscatal.7b02032
- 3 Teschner, D. et al. The roles of subsurface carbon and hydrogen in palladium-catalyzed alkyne hydrogenation. *Science* **320**, 86–89 (2008). doi:10.1126/science.1155200
- 4 Blume, R. et al. Structural and chemical properties of NiOx thin films: oxygen vacancy formation in O₂ atmosphere. *ChemPhysChem* **24**, e202300231 (2023). doi:10.1002/cphc.202300231
- 5 Freund, H. J. et al. Preparation and characterization of model catalysts: from ultrahigh vacuum to in situ conditions at the atomic dimension. *J. Catal.* **216**, 223–235 (2003). doi:10.1016/S0021-9517(02)00073-8
- 6 Armbrüster, M. et al. How to control the selectivity of palladium-based catalysts in hydrogenation reactions: the role of subsurface chemistry. *ChemCatChem* **4**, 1048–1063 (2012). doi:10.1002/cctc.201200100
- 7 Tew, M. W., Miller, J. T. & van Bokhoven, J. A. Particle size effect of hydride formation and surface hydrogen adsorption of nanosized palladium catalysts: L₃ edge vs K edge X-ray absorption spectroscopy. *J. Phys. Chem. C* **113**, 15140–15147 (2009). doi:10.1021/jp902542f
- 8 Davoli, I. et al. The local electronic structure of PdO crystal and PdO catalyst supported on SiO₂ and γ-Al₂O₃ from L₃ and L₁ x-ray absorption Pd edge in XANES spectra. *Solid State Commun.* **48**, 475–478 (1983). doi:10.1016/0038-1098(83)90857-8
- 9 Tew, M. W., Janousch, M., Huthwelker, T. & van Bokhoven, J. A. The roles of carbide and hydride in oxide-supported palladium nanoparticles for alkyne hydrogenation. *J. Catal.* **283**, 45–54 (2011). doi:10.1016/j.jcat.2011.06.025
- 10 Zhang, J. et al. Composition of the green oil in hydrogenation of acetylene over a commercial Pd-Ag/Al₂O₃ Catalyst. *Chem. Eng. Technol.* **39**, 865–873 (2016). doi:10.1002/ceat.201600020
- 11 Shirakawa, H., Ito, T. & Ikeda, S. Raman scattering and electronic spectra of poly(acetylene). *Polym. J.* **4**, 460–462 (1973). doi:10.1295/polymj.4.460

- 12 Schaffer, H., Chance, R., Silbey, R., Knoll, K. & Schrock, R. Conjugation length dependence of Raman scattering in a series of linear polyenes: implications for polyacetylene. *J. Chem. Phys.* **94**, 4161-4170 (1991). doi:10.1063/1.460649
- 13 Belov, D., Ol'khov, Y. A., Belov, G., Solovyeva, T. & Kozub, G. Thermal study of irradiated polyacetylene films. *J. Therm. Anal. Calorim.* **46**, 237-243 (1996). doi:10.1007/BF01979964
- 14 Shittu, T. D. & Ayodele, O. B. Catalysis of semihydrogenation of acetylene to ethylene: current trends, challenges, and outlook. *Front. Chem. Sci. Eng.* **16**, 1031-1059 (2022). doi:10.1007/s11705-021-2113-3
- 15 Shao, L. et al. Nanosizing intermetallic compounds onto carbon nanotubes: active and selective hydrogenation catalysts. *Angew. Chem. Int. Ed.* **50**, 10231-10235 (2011). doi:10.1002/anie.201008013
- 16 Yang, B., Burch, R., Hardacre, C., Hu, P. & Hughes, P. Mechanistic study of 1,3-butadiene formation in acetylene hydrogenation over the Pd-based catalysts using density functional calculations. *J. Phys. Chem. C* **118**, 1560-1567 (2014). doi:10.1021/jp408807c
- 17 Gladisch, H. Acetylen-herstellung im elektrischen lichtbogen. *Chem. Ing. Tech.* **41**, 204-208 (1969). doi:10.1002/cite.330410416
- 18 Chen, M. M. et al. Thermodynamics insights into the selective hydrogenation of alkynes in C₂ and C₃ streams. *Ind. Eng. Chem. Res.* **60**, 16969-16980 (2021). doi:10.1021/acs.iecr.1c03553
- 19 Shaikhutdinov, S. K. et al. Effect of carbon deposits on reactivity of supported Pd model catalysts. *Catal. Letters* **80**, 115-122 (2002). doi:10.1023/a:1015452207779
- 20 Liu, Y. et al. Adsorbate-induced structural evolution of Pd catalyst for selective hydrogenation of acetylene. *ACS Catal.* **10**, 15048-15059 (2020). doi:10.1021/acscatal.0c03897